# Research

evolution, ecology, behaviour

*Varroa destructor*, *Apis mellifera*, *Varroa* resistance, recapping, brood removal, mite infertility

**Author for correspondence:**
Isobel Grindrod
e-mail: i.r.grindrod@edu.salford.ac.uk

# Parallel evolution of *Varroa* resistance in honey bees: a common mechanism across continents?

Isobel Grindrod and Stephen J. Martin

School of Environment and Life Sciences, University of Salford, Manchester M5 4WT, UK

IG, 0000-0002-9666-7760; SJM, 0000-0001-9418-053X

The near-globally distributed ecto-parasitic mite of the *Apis mellifera* honeybee, *Varroa destructor,* has formed a lethal association with Deformed wing virus, a once rare and benign RNA virus. In concert, the two have killed millions of wild and managed colonies, particularly across the Northern Hemisphere, forcing the need for regular acaricide application to ensure colony survival. However, despite the short association (in evolutionary terms), a small but increasing number of *A. mellifera* populations across the globe have been surviving many years without any mite control methods. This long-term survival, or *Varroa* resistance, is consistently associated with the same suite of traits (recapping, brood removal and reduced mite reproduction) irrespective of location. Here we conduct an analysis of data extracted from 60 papers to illustrate how these traits connect together to explain decades of mite resistance data. We have potentially a unified understanding of natural *Varroa* resistance that will help the global industry achieve widespread miticide-free beekeeping and indicate how different honeybee populations across four continents have resolved a recent threat using the same suite of behaviours.

## 1. Introduction

Throughout the world the western honeybee, *Apis mellifera*, is an irreplaceable species, particularly in terms of their pollination services that contribute to food security and wider ecosystem health [1,2]. Despite the huge reliance on and commercialization of honeybees, their populations have for many years suffered high losses, particularly over the winter period [3,4]. While it is apparent that numerous stressors such as intensive agriculture and diseases are owing to this decline, it is well established that during the past 70 years the synergy between Deformed wing virus (DWV) and its vector *Varroa destructor* has become a critical global threat to honeybee health [5].

After *Varroa* jumped the species barrier around the 1950s, from its native host *Apis cerana* (Asian honeybee) onto *A. mellifera,* it spread globally along with DWV [6–8]. Currently only Australia and a few small, isolated islands are free of both DWV and *Varroa* [9,10]. As *A. mellifera* was completely naive to the mite, *Varroa* typically increased uncontrollably, which, coupled with a new viral transmission route (during mite feeding), led to the catastrophic collapse of both managed and feral populations across the globe [11]. As a result, particularly in the Northern Hemisphere, the constant use of acaricides is necessary for beekeeping to survive [12]. However, while acaricides help reduce the *Varroa* and DWV burden, they also remove the selective pressure from *A. mellifera* hampering any adaptation to the parasite [13–18]. Only three *Varroa*-infested *A. mellifera* populations exist without DWV and hence have never been treated with acaricides. These exist in the highlands of Papua New Guinea, the Solomon Islands [19] and on the island of Fernando de Noronha, Brazil [20]. Although the mechanism is unknown, *Varroa*

resistance arose quickly, caused no colony losses and resulted in high levels of infertile mites in the Fernando de Noronha population [20].

In the presence of DWV and the absence of treatment, *A. mellifera* populations are able to gradually develop *Varroa* resistance, typically after an initial period of colony losses [21]. Resistance is the ability of a population to survive long term without any treatment for *Varroa* within a given environment [16]. Thus, we do not view resistance as a fixed trait but the product of adaptive traits and adaptation to the local environment [17,22] in terms of the surrounding managed and feral colonies. *Varroa*-resistant colonies first appeared in Africa [23,24] and Africanized honeybees (African × European hybrid) in South America [25], and were associated with widespread lack of control due to acaricide cost and the general resilience of the bee populations. These populations, unlike in developed countries, are not frequently treated or medicated against a range of pathogens and pests [26]. Despite this, a small but increasing number of beekeepers in Europe [27], the UK [28,29] and the USA [30,31] have stopped all regular acaricide treatment and often establish their managed colonies from feral swarms [31,32].

Independently, each *Varroa*-resistant honeybee population previously studied across seven countries has developed the same traits to control the mite. These are: (i) brood removal, in which *Varroa*-infested pupae are removed; (ii) recapping, where holes are created allowing direct access to the pupa and then resealed; and (iii) mite infertility, where female mites are unable to produce viable (mated) female offspring.

Unlike many maladies the *Varroa*–DWV assoiation is a new problem especially in evolutionary terms, since *Varroa* has only been in *A. mellifera* populations between 15 and 70 years, depending on the location [6]. However, three studies [27,33,34] using the same methods found two traits (increased recapping and mite infertility) in *Varroa*-resistant populations in South Africa, Brazil, France, UK, Norway and Sweden, countries with different environmental conditions (tropical to sub-artic). This indicates that *Varroa* resistance has arisen in multiple locations, irrespective of honeybee variety or environment, especially since recapping behaviour is rarely seen in *Varroa*-naive populations in Australia, Isle of Man and Isle of Colonsay, UK [33,34].

This study's aim is to bring together data from 60 publications over the past 40 years combined with a recent breakthrough study [27] to compare the expression of brood removal, recapping and mite infertility in resistant colonies and susceptible colonies. We then construct a potential framework that links these three traits and use modelling to explore various aspects of the framework.

## 2. Methods

### (a) Data collection

We searched published literature using Scopus, Web of Science and Google Scholar to collect data on the three key traits namely brood removal, recapping and *Varroa* non-reproduction in worker brood from susceptible and resistant *A. mellifera* populations. We define resistant populations as those that have survived five or more years without any form of mite treatment, although many populations studied have survived untreated more than 10 years and some for decades. Despite many studies

used to collate the data, the methods employed are all basically the same. Furthermore, a study was only included if a minimal sample size of 50 cells was recorded, used natural comb, and only included cells infested with a single foundress.

We extracted information from 60 key data-rich papers. Where possible single colony data were extracted. For example, all recapping data ($n = 163$) came from single colonies; for brood removal nine of the 86 data points are colony averages and for mite infertility 75 of the 99 data points are colony averages, due to sample size limitations (see electronic supplementary material data for all source data). No susceptible colonies are known from where Africanized and African bees occur hence comparisons with resistant colonies in these locations are not possible. Almost all the data collected concerns the Korean 'K' haplotype of *Varroa* (see electronic supplementary material data for more information).

### (b) Brood removal

We used the standard bee search string (Apis mellifera OR honeybee OR honeybee) AND (removal OR brood removal OR hygienic behaviour OR VSH OR varroa-sensitive hygiene OR varroa-specific hygiene) AND varroa. We looked for studies that measured the removal of brood that had been artificially or naturally infested (one study [35]) with *Varroa*. Studies using artificial infestation all had to follow the same basic protocol outlined in [33]. In brief, a frame of freshly capped brood is taken from a colony and mites are inserted carefully into the capped cells containing recently capped cells. After around 10 days in the colony the frame is inspected, and the number of infested cells removed is recorded.

### (c) Recapping

We used the standard bee string AND (re-capping OR recapping) AND varroa. To be included, studies had to have measured the recapping of *Varroa*-infested cells following the correct protocol outlined in [36,37].

### (d) Mite infertility

We used the standard bee search string AND (varroa OR varroa mite OR mite) AND (reproduction OR non-reproduction OR fertility OR infertility). Here we define infertility as the inability to produce a viable (mated) female offspring and so we collected data following this definition. Importantly, some data used were collected from papers that used the definition of no egg laying. The justification for this is that non-egg laying also falls within the definition, and at worst provides an underestimate of the reduced reproductive rate of mites. To calculate the effect of brood removal on offspring production by *Varroa*, a simple equation was formulated,

$$(1 - a) \times b = c,$$

where $a$ is proportion of infested cells removed, $b$ is maximum number of viable offspring produced per cycle and $c$ is average number of viable female offspring produced per reproductive cycle.

### (e) Data analysis

The sample sizes (in cells) were used to calculated weighted averages for each of the traits for resistant and susceptible populations. Statistical analyses were conducted in Minitab version 18 on unweighted data [38]. Mann–Whitney $U$-tests were used to compare the removal abilities, recapping abilities and infertile mite proportions of resistant and susceptible populations. Statistical significance for all tests was $p < 0.05$.

The effect of brood removal on mite and honeybee population growth was modelled using the BEEHAVE model [39]. Increasing worker pupal mortality rates were used to simulate brood removal (as dead brood is removed in the simulation). The mortality was independent of mite infestation as the effect of DWV was removed from the equation for simplicity since within the BEEHAVE model DWV also affects pupa mortality confounding the observation of the effect of brood removal. This simplification was deemed acceptable as the result would only provide an underrepresentation. In actuality, as bees target infested cells it would likely take less removal to achieve the same outcome.

### (f) Framework construction
After collecting and analysing the data, we constructed a hypothetical framework to explain how many of the various traits are connected. Data from this study or findings from related studies were used to justify the proposed link between each trait.

## 3. Results

### (a) Honeybee behaviour
Recapping behaviour is the resealing of holes made in the cap that covers the developing worker pupa, holes allow better access to the signal(s) that trigger hygienic behaviour [33,40]. We collected data from 163 colonies from five studies that took place across seven countries (figure 1$c$). This showed that in resistant colonies significantly more infested cells are recapped than in susceptible colonies (55% versus 33%) ($U = 1280$, $p < 0.00001$).

Brood removal is a trait of honeybees where diseased or dead pupae are removed. It defends the colony against the spread of several diseases including chalkbrood, American foul brood and *Varroa* infestation. Data from mite-infestation experiments from 403 colonies (86 data points) across 10 studies conducted in seven countries demonstrate that resistant colonies are significantly ($U = 341.5$, $p < 0.0001$) better at removing mite-infested brood than susceptible colonies (38% versus 22%) (figure 1$b$). When separated into populations both Africanized bees and their African relatives (*A. m scutellata* and *A. m capensis*) have significantly greater ($U = 83$, $p < 0.0001$ and $U = 207.5$, $p = 0.002$) removal abilities than susceptible colonies in Europe.

### (b) *Varroa* reproduction
We used the equation '$(1 - a) \times b = c$' (see Methods), which generates a linear relationship between brood removal and reproductive output (figure 1$d$). The removal of 38% and 22% infested brood in resistant or susceptible colonies (figure 1$b$) predicts 0.87 (resistant) and 1.09 (susceptible) viable female offspring are produced per reproductive cycle when no removal allows 1.4 viable female offspring to be produced [42]. If a maximum value of 1.6 (56) is used, values of 0.99 (resistant) and 1.25 (susceptible) are obtained. These values are independent of the total number of reproductive cycles performed, which varies between two and three [43–45]. The decrease in reproductive output increases the proportion of infertile mites (see Discussion for details). Data from 786 colonies (99 data points) across 40 studies in 14 countries showed that resistant populations had

significantly ($U = 28$, $p < 0.0001$) greater proportions of infertile mites than susceptible colonies (45% versus 17%) (figure 1$e$).

### (c) Colony level effects
The BEEHAVE model predicted that removing greater than 40% of infested pupae results in negative mite population growth (figure 1$f$). Additionally, it predicted that, irrespective of infestation status, if the brood removal rate were to exceed 40% in spring, 55% in summer or 60% in winter, the colony would collapse (electronic supplementary material, figure S2). However, resistant colonies now typically only have worker-brood infestation rates of around 4% (figure 1$h$).

### (d) Decreasing worker-brood infestation levels
In the Africanized colonies, which are all resistant, average worker-brood infestation rates have fallen from 20% during 1996–1998 to 4% in 2018–2019 (figure 1$h$). Additional preliminary data from UK-resistant colonies ($n = 44$) collected by the authors and [46] found that brood infestation averaged at 6% and was not significantly different to Africanized colonies in 2018/2019 ($U = 460$, $p = 0.052$).

### (e) Framework
Using the data and analyses presented above, we constructed a framework to link them together to explain how *Varroa* resistance may develop in *A. mellifera* (figure 1). Our interpretation centres on the idea that an existing trait, hygienic behaviour, when adapted to detecting and removing mite-infested pupae, can explain all other traits. Given the data and the models used as well as the findings of other studies, we believe our framework to be the most plausible interpretation of the results we have presented here. Further justifications for the framework are presented in the discussion.

## 4. Discussion
The proposed framework attempts to explain how *Varroa* resistance may develop in honeybee (*A. mellifera*) populations. The framework suggests that resistance is a sequence of events that generate the key traits (increased recapping, brood removal and mite infertility) rather than a single trait [21,47]. Here we found that the enhanced expression of these three key traits is common among resistant populations. This independent occurrence of the key traits within colonies across the world could be an example of parallel evolution [27], because while the recapping and removal behaviours predate *Varroa*, they have been co-opted to control *Varroa*, recapping is rare trait in mite-naive colonies, but occurs at low and high levels in susceptible and resistant colonies respectively [33,40]. Similarly, other traits such as brood suppression of mite reproduction [48], or DWV tolerance [49,50] may complement those within the framework. There is also likely to be a mite element to resistance which could be illuminated by further studies into the coevolution of *A. mellifera* and *Varroa* [51,52]. As resistance is a population level trait rather than a single colony trait, a resistant colony becomes vulnerable if moved out of its population and could collapse if a sudden influx of mites occurs due to excessive (40–60%) brood removal (electronic supplementary material, figure

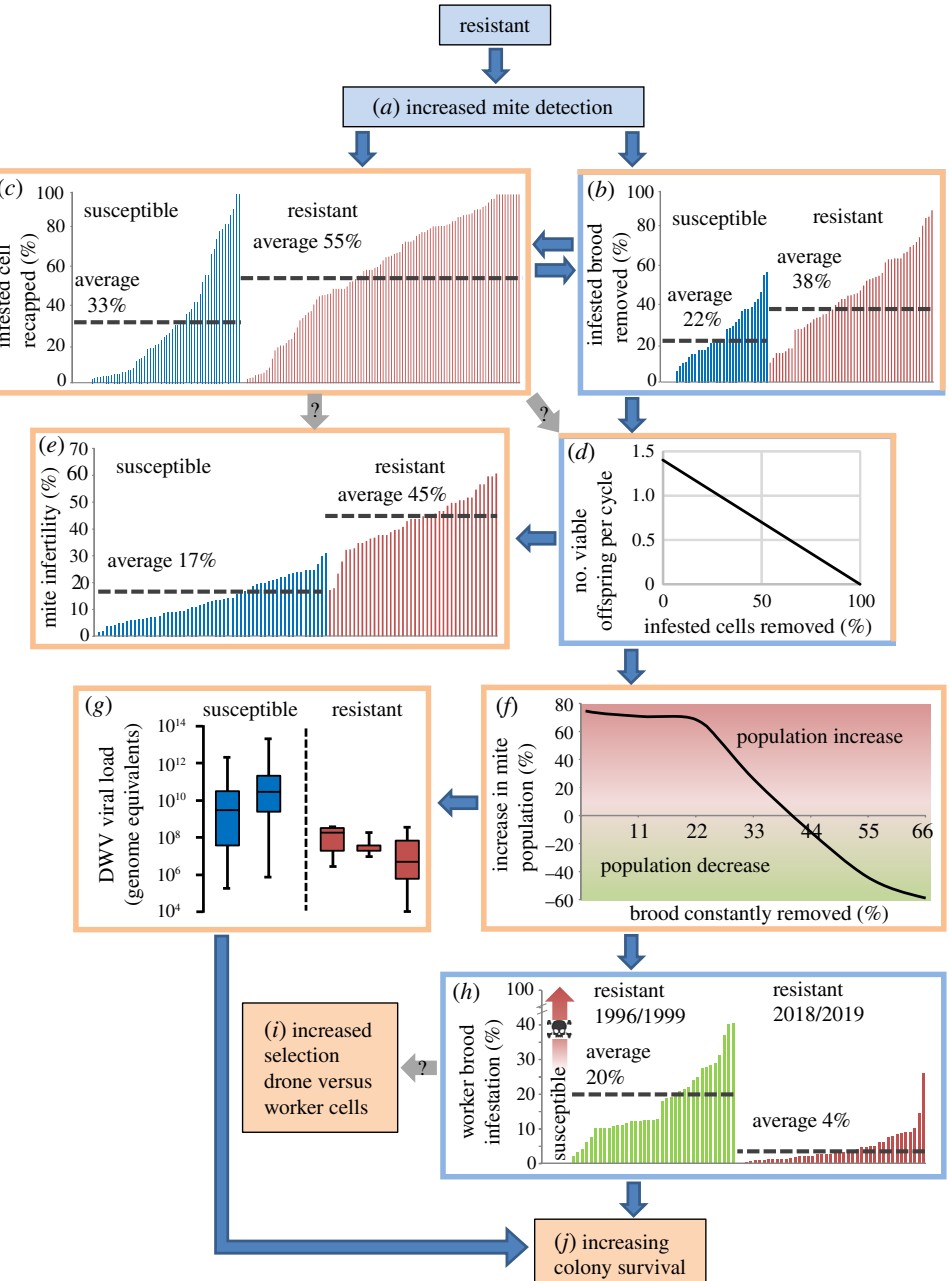

**Figure 1.** A proposed framework for the development of *Varroa* resistance. Boxes in blue (*a*) or with a blue border (*b,d,h*) are 'causes' of the 'effects' that are indicated by boxes in orange (*i,j*) or with orange borders (*b–g*). All source data for each chart are available in the electronic supplementary material, tables S1–S8 and figure S1. Arrows with a question mark indicate possible links suggested in the literature. In box *h*, the red arrow indicates that in untreated, susceptible colonies *Varroa* infestations continuously rise until colony death. Deformed wing virus data in box *g* are adapted from [41] and discussed below. (Online version in colour.)

S2). This may explain why resistant colonies moved out of their population typically do not survive [53] (S.J.M. 2017, 2019, personal observation).

## (a) Honeybee behaviour

The framework begins with the increased detection of *Varroa*-infested cells, an ability that has been linked to resistant bees by numerous studies [33,54–57] (figure 1*a*). Unlike most brood diseases *Varroa*-DWV is a chronic condition that does not kill the developing host pupae but shortens its lifespan as an adult [43,58,59]. Bees already have a well-developed hygienic behaviour response but it typically deals with diseases that cause dead brood [60]. Despite this, clear evidence exists for the detection of infested cells, directly from six mite insertion experiments and one natural

infestation experiment (figure 1*b*), and indirectly from the behaviour known as recapping (figure 1*c*).

The fact that on average resistant colonies remove and recap significantly greater proportions of infested cells than susceptible colonies (figure 1*b,c*) indicates that increased detection of infested cells causes these traits to increase. Additionally, recapping has been shown to be positively correlated with brood removal [27,33] further suggesting a common trigger. Increased recapping may occur because more sensitive adults [55–57] investigate sealed brood around infested cells either due to a diffuse signal emanating from infested cells or increased cursory checking near infested cells [33,40].

Typically, hygienic behaviour tests use the freeze-killed brood method [61], and this does not correlate with removal of mite-infested brood [33,46,62–66]. However, this does not

negate the contribution of hygienic behaviour to mite resistance, since the cues are different (living versus dead pupae) [47] and freezing kills many brood at the same time in the same location, thus generating an abnormally high concentration of cues. Therefore, if colonies perform exceptionally well (remove greater than 95% dead brood within 24 h) they may remove a reasonable amount (average of 66%) of *Varroa*-infested brood and have high recapping rates [65].

It is unclear whether the cues involved are emanating from the mites or pupae [55,56,67–69] or both [54], since parasitization by *Varroa* and DWV infection causes changes to the chemical profile of pupae [68,69,70–73]. Six compounds (four ketones and two acetates) have been detected on both infested pupae and mites, and although all adult workers can detect these compounds only workers from resistant colonies can distinguish the mix of six compounds from healthy brood [54]. Other studies [71,74] have detected different compounds that could also stimulate a hygienic response. The general consensus is that multiple chemical cues are involved in hygienic behaviour, which may prevent the loss of healthy brood, if a cell is wrongly opened the subsequent lack of the secondary cue could trigger resealing or 'recapping' [40]. Indeed, recapping of both non-infested and infested cells is consistently elevated in all resistant populations [33]. The hole made in the cell cap is generally less than 1 mm in non-infested cells, but significantly larger (up to 5 mm) in infested cells [33,46], which may increase the detection of less volatile cues such as those described [54].

## (b) *Varroa* reproduction

In our framework, we link increased removal of mite-infested to reduced reproductive output and thus increased mite infertility (figure 1b,d,e). Previous studies have also suggested links between increased brood removal, potentially recapping [27,75] and reduced mite reproductive success [76]. In agreement, we found that resistant colonies had a significantly greater percentage of infertile mites (figure 1e). A simple explanation is that disrupting the very uniform sequence of mite-reproduction leads to foundress-mites producing less offspring and depleting their finite supply of 18–30 eggs [77–80] and limited supply of spermatozoa [80,81]. Infertile mites have fewer spermatozoa [82], and the number of laid eggs steadily declines in mites preforming more than two reproductive cycles [77]. Using the simple equation (figure 1d), the estimated reproductive values for resistant and susceptible colonies of between 0.87–0.99 and 1.09–1.25, respectively, were similar to actual values from resistant and susceptible colonies [27,33,83]. Whatever the reason, the reproductive asynchrony caused by the removal of infested pupa causes less mites to contribute to the next generation, thus population growth slows and there is a reduced proportion of new fertile mites compared to older infertile mites [76,84]. In addition to brood removal, reductions in mite fertility may be the result of similar interruptions by recapping [75] and/or brood effects [48], but more data is needed.

## (c) Colony-level effects

Reduced fertility we then linked to reduced population growth because our BEEHAVE model predicted that infested brood removal above 40% caused negative mite population growth (figure 1f). Thus, in our framework the detection and removal via cannibalization of infested worker-brood leads to reduced mite population growth, a commonly occurring outcome in surviving populations [47]. Additionally, because brood removal varies within a population (figure 1b) the BEEHAVE model helps explain the fluctuating mite populations observed in long-term studies of resistant colonies [83,85,86]. Other studies also found an association between increased mite infertility and a reduced mite burden [24,63,87–89], again suggesting it may link brood removal and population growth.

Furthermore, reduced mite burden also reduces the number of viral vectors [22], causing lower viral titres (figure 1g) [41,90–92] and a reduced number of deformed bees [93–95]. One study [96] found that removal above 95% of freeze-killed pupae lowered mite population growth and significantly lowered DWV titres in workers than colonies below 95% removal. However, cannibalism of infested pupae allows DWV prevalence to remain high [97] even in resistant populations [98], but titres fall since oral (natural) viral transmission is much less infective than via injection [93,97].

## (d) Decreasing worker-brood infestation levels

In non-resistant untreated colonies, mite populations increase until colony collapse with increasing brood infestation levels from 30% to 100% at colony collapse [99], whereas in resistant colonies worker-brood infestation rate is maintained below 20% (figure 1h). Interestingly, we found that worker-brood infestation has fallen significantly ($U = 123$, $p < 0.0001$) from 20% to just 4% over the past two decades in resistant colonies in South America (figure 1h) currently the only location with long-term data.

We speculate that this is because mites are increasingly waiting for drone brood, which is not targeted by hygienic behaviour in either *A. mellifera* nor *A. cerana* [100]. Furthermore, the proportion of mites on adult bees decreased when drone brood was plentiful and increased when it was scarce [101]. Similarly, in resistant colonies from Uruguay the ratio of the mites' distribution between worker and drone cells was much greater (1 : 12.6) than in susceptible colonies (1 : 5.7) [102]. Heavily infested drone brood has also been observed in resistant populations in Mexico, Brazil and South Africa (30% [33]); however, much of the evidence is anecdotal and needs studying further.

In fact, the evolutionary reason why *V. jacobsoni* avoids worker brood in its natural host *A. cerana* remains unclear. It is well established in *A. cerana* that *V. jacobsoni* rarely reproduces in worker brood [103–106], and the drone pupa dies if infested by multiple mite families and becomes entombed within the cell rather than removed [107]. When *V. destructor* mites are artificially inserted into incubated *A. cerana* worker brood 30–50% of the pupae die [108], potentially due to a saliva toxin protein from *V. destructor*, but no mortality occurs in *A. mellifera* [108,109]. This implies that hygienic behaviour in *A. cerana* relies on detecting dead brood, making the ability to detect living infested pupa and mites [54] in *A. mellifera* even more unique. However, further studies in *A. cerana* are required to differentiate between or link together (i) the detection and removal of living mite-infested brood, (ii) social apoptosis and removal of dead brood, and (iii) any coevolution by *Varroa* or worker brood that prevents mite reproduction.

Finally, in a small resistant *A. mellifera* population on the remote Fernando de Noronha Island, Brazil, adult mite infestation levels fell from 26% in 1991 to 1–2% in 2016. However, in worker and drone brood infestation levels have stabilized around 20% and 40%, respectively (electronic supplementary material, figure S3) [20,110], despite very high infertility rates [20]. This may be explained by the very rare absence of DWV from this population that allows high brood infestation levels to persist without the negative impacts of DWV. Confirmatory studies from the other two DWV-free *Varroa*-infested populations [19] are needed.

## (e) Reduced colony losses

The final link in our framework is that reduced mite and virus burden will lead to enhanced colony survival [43]. Indeed, the reduction of mite burden and associated enhanced survival is the primary function of acaricides. Enhanced survival is hard to measure as susceptible colonies are usually treated with acaricides. However, the annual loss rates of treated colonies are higher than resistant populations in Le Mans and Avignon (France) [111]. Additionally, over 100 beekeepers across a 2500 km² region of north Wales (UK) have maintained 499 colonies treatment free for 11 years [32], and in Swindon (UK) a small beekeeper group have kept treatment-free colonies since 1995 [112], and neither group have report increased losses. In South Africa, after an initial period of high losses, annual colony losses stabilized at around 5% between 1998 and 2004, which is similar to pre-*Varroa* levels [23]. Also, in Algeria, Tunisia and Morocco, initial colony losses were high, although short-lived [113]. Across most of Africa [23,113–116] and in Africanized colonies throughout Latin America no widespread losses were reported where lack of acaracide use, due to cost and availability, may have helped resistance develop. Instead, widespread colony losses occurred in the Northern Hemisphere as *Varroa* spread from Asia throughout Europe and into the Americas, where acaracides were quickly adopted.

## (f) Variability of data

A substantial issue when it comes to measuring resistance traits is the inherent variability within colonies and thus across populations. Within a colony, traits themselves are not static and fluctuate with the changing season along with the associated availability of worker and drone brood and the infestation level [52,117–123]. Variability is also likely due to temporal changes in the composition of the different hygienic workers. To elaborate, the three main stages of brood removal (the initial detection and opening of the cell cap, the full uncapping of the cell and finally removing or cannibalizing the pupae or recapping the cell [124]) are conducted by bees of different ages and sensory acuity, a division of labour further affected by genetic, neural, social and environmental conditions [55,125–128]. For example, an imbalance of 'uncapper' versus 'recapper' bees may cause many brood cells to be left open [55]. Consequently, it can be very hard to accurately measure resistance-associated traits [117,118,129], resulting in a high degree of variability within colonies and across colony-level datasets (figure 1*b,c,e*). Ultimately, variability severely affects selection programmes (reviewed in [130]), whereas in natural selection-based experiments such as bond experiments [15] and black box experiments [13,131], assumptions on the importance of traits are not made.

## 5. Conclusion

This study shows that the resistance traits of recapping, brood removal and mite infertility are expressed at significantly higher levels in resistant colonies than susceptible ones, and we present a framework to potentially explain how these common traits shared by resistant colonies can link together. Although many local sub-species exist, *A. mellifera* remains a single species and environmental conditions within the colony (i.e. those that *Varroa* are subject to) remain remarkably constant irrespective of location, which has aided its semi-domestication and global distribution. Natural bee-driven resistance to *Varroa* is a sustainable, long-term solution, prevents the constant usage of acaricides, will not weaken bees to any other maladies should they arise and may provide an example of parallel evolution with the same three traits arising in populations in several different continents.

Data accessibility. Data available from the Dryad Digital Repository: https://doi.org/10.5061/dryad.h18931zk6p [132]. The data are provided in electronic supplementary material [133].

Authors' contributions. I.G.: conceptualization, data curation, formal analysis, methodology, writing-original draft, writing-review and editing; S.J.M.: conceptualization, supervision, writing-original draft, writing-review and editing.
Both authors gave final approval for publication and agreed to be held accountable for the work performed therein.

Competing interests. The authors declare no competing interests.

Funding. Funding for I.G. was provided by Bee Diseases Insurance Ltd, UK.

Acknowledgements. We would like to thank four reviewers and the editor for helping to improve the manuscript, and M. Spivak and M. Oddie for initial discussions.

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
