## [Peer Review File · Proceedings of the Royal Society B: Biological Sciences]

Review History

RSPB-2020-2795.R0 (Original submission)

Review form: Reviewer 1

Recommendation

Accept with minor revision (please list in comments)

Scientific importance: Is the manuscript an original and important contribution to its field?

Excellent

General interest: Is the paper of sufficient general interest?

Excellent

Quality of the paper: Is the overall quality of the paper suitable?

Excellent

Is the length of the paper justified?

Yes

Should the paper be seen by a specialist statistical reviewer?

No

Do you have any concerns about statistical analyses in this paper? If so, please specify them explicitly in your report.

No

It is a condition of publication that authors make their supporting data, code and materials available - either as supplementary material or hosted in an external repository. Please rate, if applicable, the supporting data on the following criteria.

Is it accessible?

Yes

Is it clear?

Yes

Is it adequate?

Yes

Do you have any ethical concerns with this paper?

No

Comments to the Author

This is an excellent synthesis of prior research and new findings on honey bee mechanisms of resistance to *Varroa destructor* mites. Figure 1 and especially Fig 2 are brilliant. The take-home message for me is that it doesn't take a lot of one kind of behavior (e.g., average of 46% removal of mite-infested brood, 61% recapping) to lead to reduced mite fertility, reduced mite load, viral load and colony survival. The selection for mites preferring drone brood over worker brood due to the bees surveillance and disruption of mite-infested worker brood (and no surveillance or disruption of drone brood) makes a lot of sense, but requires more future study.

My only bone to pick is about the word "learning" (lines 101, 104, 200), and the assumption that uncapping cells is a rare trait (lines 120, 203). Honey bees uncap diseased brood: they uncap (but don't always remove) some portion of brood infected with chalkbrood, AFB, virus infected brood, and they uncap cells that have wax moth larvae tunneling under the wax. Uncapping and recapping are traits that predate *Varroa* in *Apis mellifera*. If a population of colonies has little or no disease, it might appear that uncapping/ recapping is rare. But from my experience, the most unhygienic of colonies uncap diseased brood – the problem occurs if they try to remove the brood when infectious, or even if they leave the infected brood in place and other bees come into contact with it, the uncapping/ removal increases the risk of pathogen transmission. My point is that I don't think uncapping and recapping are learned behaviors; they are part of the honey bee behavioral repertoire, but genetically some colonies are more prone to detect and uncap more quickly than other colonies. That said, I do think some learning may occur within a colony that is infected with a pathogen or infested with mites. Once cells have been opened, other bees may become "primed" or sensitized to the olfactory stimulus, but to my knowledge this has not been tested.

Very minor points:

Line 151: should say 40% of the infested pupae?

Line 213: tasks are performed by bees (plural) of different ages...

Nice work.

Review form: Reviewer 2 (Per Kryger)

Recommendation

Major revision is needed (please make suggestions in comments)

Scientific importance: Is the manuscript an original and important contribution to its field?

Good

General interest: Is the paper of sufficient general interest?

Good

Quality of the paper: Is the overall quality of the paper suitable?

Good

Is the length of the paper justified?

Yes

Should the paper be seen by a specialist statistical reviewer?

No

Do you have any concerns about statistical analyses in this paper? If so, please specify them explicitly in your report.

No

It is a condition of publication that authors make their supporting data, code and materials available - either as supplementary material or hosted in an external repository. Please rate, if applicable, the supporting data on the following criteria.

Is it accessible?

Yes

Is it clear?

Yes

Is it adequate?

Yes

Do you have any ethical concerns with this paper?

No

Comments to the Author

The manuscript is based on the compilation of the data, I think, from all available literature published over the past several decades. As such, the main novelty is in the meta-analysis of the resulting massive dataset. It creates new learnings from previous publications, which is applaudable.

I have a few observations, that I think could improve the manuscript. I am always disturbed by the use of not generally accepted acronyms. In the current publication, NVR FKB and AHB are used, the latter two for Freeze killed brood respectively Africanized honey bees only occurs few times, and really does not require abbreviation. NVR stands in for naturally varroa resistant. I have two issues with this. I can't be expected to learn a set of new acronyms for each paper I read, and it really reduces the readability for the casual reader, that just wants to gloss over the paper in 10 weeks or 20 years time, they may just put the paper down, as being unreadable. More important than my discomfort, in most parts of the paper, naturally varroa resistant bees are discussed as opposed to susceptible bees. So the logic question is, are susceptible bees unnatural? Obviously not. They are susceptible whereas the other bees are resistant. Only in the second large

paragraph, L267 to L276 are bees discussed, that have been bred for high level of hygienic behaviour, but does that really qualify as unnatural?

Figure 1 and figure 2 are having a very similar message, I think figure 2 is sufficient, maybe it could be improved by adding blue and orange borders around the figures, as in figure 1.

L113-4 instead the comma and the next sentence, I suggest you write: Europe. Comparisons to susceptible colonies where Africanized and African bees occur is impossible, as these haven't be found in time to be assessed.

Paragraph 163-171. The reduction of DWV that results from resistant bees, is here only described as a general lower titre. The prevalence of DWV sick bees probably is reduced too, inside the colony, which has been shown relevant for survival (Dainat B, Neumann P. Clinical signs of deformed wing virus infection are predictive markers for honey bee colony losses. *Journal of invertebrate pathology*. 2013 Mar 1;112(3):278-80 and Francis RM, Nielsen SL, Kryger P. Varroa-virus interaction in collapsing honey bee colonies. *PloS one*. 2013 Mar 19;8(3):e57540.).

This brings up another issue, that the authors could reflect more on. The graphs in figure 2 show a considerable level of variation, that by far exceeds the differences in means between susceptible and resistant bees. Partly this may be due to biological variation, however, as was recently shown, it is very hard to accurately measure hygienic behaviour (Mondet F, Parejo M, Meixner MD, Costa C, Kryger P, Andonov S, Servin B, Basso B, Bieńkowska M, Bigio G, Căuia E.

Evaluation of Suppressed Mite Reproduction (SMR) Reveals Potential for Varroa Resistance in European Honey Bees (*Apis mellifera* L.). *Insects*. 2020 Sep;11(9):595.). I think the problem of measurements is an important discussion, and given the level of variability in the published dataset, that could deserve a paragraph in the results part of the manuscript.

L257-259 I am lost. The effect of what is meant by its? 1 st sentence is unclear. 2nd sentence, I assume the frequency of mite infertility has been recognised by science, as a key element in understanding resistance. If it was long established, we would not have a problem. There exist many explanations, still exist nor were, I think, while I agree that the simplest is often the best.

L276 ..drone brood. Why is not so clear, any observations regarding hygienic behaviour towards drone cells?

Minor suggestions.

L9-10 First sentence of abstract can be deleted, is repeated in introduction.

L39 ..allowed for.., I suggest, ..frequently resulted in..

L104 adults, better known as: worker bees

L117 ..that were created.., I suggest ..that appear to be created..

L120 (35) un-superscript

L159 This, I assume this refers to the influx of mites, rather than the removal rate, however that is one sentence removed.

L182 ..as found, replace with ..similar to that found.. and add a quote at end of sentence.

L196 replace ..losses.. with ..loss rates..

L197 less or no greater than the losses(?) replace with ..below those .. and add .. in France (44)

L200 replacing ..single.. with ...complex... seems very appropriate

L213 replace .. is.. with ..may be..

L233 replace ..all NVR.. with all examined resistant..

L270 replace ..selected.. with bred

Review form: Reviewer 3

Recommendation

Major revision is needed (please make suggestions in comments)

Scientific importance: Is the manuscript an original and important contribution to its field?

Acceptable

General interest: Is the paper of sufficient general interest?

Acceptable

Quality of the paper: Is the overall quality of the paper suitable?

Acceptable

Is the length of the paper justified?

Yes

Should the paper be seen by a specialist statistical reviewer?

Yes

Do you have any concerns about statistical analyses in this paper? If so, please specify them explicitly in your report.

No

It is a condition of publication that authors make their supporting data, code and materials available - either as supplementary material or hosted in an external repository. Please rate, if applicable, the supporting data on the following criteria.

Is it accessible?

Yes

Is it clear?

Yes

Is it adequate?

No

Do you have any ethical concerns with this paper?

No

Comments to the Author

This is an interesting approach and a huge piece of work with the aim to use raw data from about 50 publications on Varroa resistance traits like VSH, SMR, recapping and others. However, I have some major concerns with the approach, the use of the available data and the conclusions:

1. The criteria for the natural Varroa resistant (NVR) colonies are not clearly defined. There are many data from colonies in South and Central America (Africanized honey bees) and Africa, mainly tropical regions with a huge population of wild-living honey bees. From the temperate regions there exist mostly small "populations", often not really stable over a long time period and/ or often not really defined in their structure and management.

At least it must be listed which populations are considered as NVR (and why).

2. A clear quality control of the data is lacking (or at least not described). Meanwhile there are so many papers on Varroa tolerance/ resistance that this is necessary. Furthermore it is problematic to merge data from publications of nearly 4 decades. For instance in Line 137 - 141 it is mentioned that there are two ways of measuring "infertility" and that only the definition "no viable female offspring" is used. According to the suppl. material there are definitively data included that used the parameter "no egg laying at all". This may not have a huge impact on the difference between susceptible and NVR but demonstrates the general problem with "old" data, often not even properly described in the method section. The "bee book" was published exactly to overcome this problem, in the year 2013.

In addition, Fig. 2 obviously presents data of single colonies? What about published data that are averages of several colonies without providing the single colony data? There are definitively

such data in the literature of the suppl. mat. involved.

3. The data are biased because there are many data from a few authors (see suppl. material), many unpublished data.

4. I miss a critical discussion of the real result (which are presented more or less in Fig. 2). The discussion rather represents a solid review of the current discussion on the most important resistance traits.

5. The main question that are discussed in this manuscript is whether hygienic behaviour can explain also some other traits like SMR (suppressed mite reproduction). I agree that this is currently a "hot debate" (there exist already some reviews and opinion letters) and is important for the question which approach is suitable for the selection of resistant colonies. The authors obviously try to prove their assumption "that NVR colonies, regardless of location, have all developed the same strategy to control the mite". In this discussion, some studies are not included giving a differentiated view on the complex host-parasite relationship.

Some detailed comments:

1. L 57ff: This is overstated. In fact there are extremely few beekeeper stopping treatments (promoting natural selection) and most of them lose their colonies. Importantly, there are hardly good data from such activities.

2. L68: Why should this paper (a good and interesting paper mainly on recapping) be a "breakthrough"?

3. L108ff: It should be added for each parameter/ trait the number of colonies/ different locations which are used in the data set. It is too difficult to extract this from suppl. mat.

4. L131: Reproductive cycles between 2 and 3? There are other data from Fries (1996).

5. L136 (and L260ff): "increase in the number of infertile mites as they exhaust their limited supply of eggs". Here only two papers from 1984 + 1986 are cited. To my knowledge there exist no study confirming that the lack of oocytes in the Varroa female is a reason for infertility.

6. L148: Why is DWV removed from the equation?

7. L158: At least in temperate regions the critical infestation rates are quite lower. Are these statements supported by studies?

8. L177ff: These statements are not supported by literature!

9. L197: This generalized statement is only supported by one study on a small population in Southern France.

10. L1200ff: This statement must be explained and justified through the results.

11. L 267-276: These important statements are only justified by two papers from 2002 (one dealing with AFB)

12. L278ff: This statement is not supported by the data or the cited literature!

13. Some important papers/aspects are missing in the discussion, for instance:

Eynard et al., (2020) assuming that VSH and SMR are not linked!

Schöning et al. (2012): Removal of brood/ mites elicited by DWV symptoms.

Page et al. (2016, and others): "altruistic suicide" in *A. cerana* as an important trait of the original host.

Decision letter (RSPB-2020-2795.R0)

19-Jan-2021

Dear Miss Grindrod:

I am writing to inform you that your manuscript RSPB-2020-2795 entitled "Parallel Evolution of Varroa Resistance in honey bees; a common mechanism across continents" has, in its current form, been rejected for publication in Proceedings B.

This action has been taken on the advice of referees, who have recommended that substantial revisions are necessary. With this in mind we would be happy to consider a resubmission, provided the comments of the referees are fully addressed. However please note that this is not a provisional acceptance.

Sincerely,
Dr Locke Rowe
mailto: proceedingsb@royalsociety.org

Associate Editor
Board Member: 1
Comments to Author:

Thank you for submitting this interesting paper, which has now been reviewed by 3 experts in the field. As you can see from their reviews, all the reviewers found value in your work, or the question you are addressing. However, they also raise a number of issues, particularly the 3rd reviewer. If you submit a revised manuscript, please make sure to address each of these points in both a cover letter and in the revised manuscript. While I do not like to single out reviewers for particular attention, reviewer 3 in particular has specific concerns about the value and novelty of your work, and these need to be convincingly addressed.

Reviewer(s)' Comments to Author:
Referee: 1

Comments to the Author(s)

This is an excellent synthesis of prior research and new findings on honey bee mechanisms of resistance to Varroa destructor mites. Figure 1 and especially Fig 2 are brilliant. The take-home message for me is that it doesn't take a lot of one kind of behavior (e.g., average of 46% removal of mite-infested brood, 61% recapping) to lead to reduced mite fertility, reduced mite load, viral load and colony survival. The selection for mites preferring drone brood over worker brood due

to the bees surveillance and disruption of mite-infested worker brood (and no surveillance or disruption of drone brood) makes a lot of sense, but requires more future study.

My only bone to pick is about the word “learning” (lines 101, 104, 200), and the assumption that uncapping cells is a rare trait (lines 120, 203). Honey bees uncap diseased brood: they uncap (but don’t always remove) some portion of brood infected with chalkbrood, AFB, virus infected brood, and they uncap cells that have wax moth larvae tunneling under the wax. Uncapping and recapping are traits that predate *Varroa* in *Apis mellifera*. If a population of colonies has little or no disease, it might appear that uncapping/ recapping is rare. But from my experience, the most unhygienic of colonies uncap diseased brood – the problem occurs if they try to remove the brood when infectious, or even if they leave the infected brood in place and other bees come into contact with it, the uncapping/ removal increases the risk of pathogen transmission. My point is that I don’t think uncapping and recapping are learned behaviors; they are part of the honey bee behavioral repertoire, but genetically some colonies are more prone to detect and uncap more quickly than other colonies. That said, I do think some learning may occur within a colony that is infected with a pathogen or infested with mites. Once cells have been opened, other bees may become “primed” or sensitized to the olfactory stimulus, but to my knowledge this has not been tested.

Very minor points:

Line 151: should say 40% of the infested pupae?

Line 213: tasks are performed by bees (plural) of different ages...

Nice work.

Referee: 2

Comments to the Author(s)

The manuscript is based on the compilation of the data, I think, from all available literature published over the past several decades. As such, the main novelty is in the meta-analysis of the resulting massive dataset. It creates new learnings from previous publications, which is applaudable.

I have a few observations, that I think could improve the manuscript. I am always disturbed by the use of not generally accepted acronyms. In the current publication, NVR FKB and AHB are used, the latter two for Freeze killed brood respectively Africanized honey bees only occurs few times, and really does not require abbreviation. NVR stands in for naturally varroa resistant. I have two issues with this. I can't be expected to learn a set of new acronyms for each paper I read, and it really reduces the readability for the casual reader, that just wants to gloss over the paper in 10 weeks or 20 years time, they may just put the paper down, as being unreadable. More important than my discomfort, in most parts of the paper, naturally varroa resistant bees are discussed as opposed to susceptible bees. So the logic question is, are susceptible bees unnatural? Obviously not. They are susceptible whereas the other bees are resistant. Only in the second large paragraph, L267 to L276 are bees discussed, that have been bred for high level of hygienic behaviour, but does that really qualify as unnatural?

Figure 1 and figure 2 are having a very similar message, I think figure 2 is sufficient, maybe it could be improved by adding blue and orange borders around the figures, as in figure 1.

L113-4 instead of the comma and the next sentence, I suggest you write: Europe. Comparisons to susceptible colonies where Africanized and African bees occur is impossible, as these haven't been found in time to be assessed.

Paragraph 163-171. The reduction of DWV that results from resistant bees, is here only described as a general lower titre. The prevalence of DWV sick bees probably is reduced too, inside the colony, which has been shown relevant for survival (Dainat B, Neumann P. Clinical signs of deformed wing virus infection are predictive markers for honey bee colony losses. *Journal of invertebrate pathology*. 2013 Mar 1;112(3):278-80 and Francis RM, Nielsen SL, Kryger P. *Varroa-virus interaction in collapsing honey bee colonies*. *PLoS one*. 2013 Mar 19;8(3):e57540.).

This brings up another issue, that the authors could reflect more on. The graphs in figure 2 show a considerable level of variation, that by far exceeds the differences in means between susceptible and resistant bees. Partly this may be due to biological variation, however, as was recently shown, it is very hard to accurately measure hygienic behaviour (Mondet F, Parejo M, Meixner MD, Costa C, Kryger P, Andonov S, Servin B, Basso B, Bienkowska M, Bigio G, Căuia E.

Evaluation of Suppressed Mite Reproduction (SMR) Reveals Potential for Varroa Resistance in European Honey Bees (*Apis mellifera* L.). *Insects*. 2020 Sep;11(9):595.). I think the problem of measurements is an important discussion, and given the level of variability in the published dataset, that could deserve a paragraph in the results part of the manuscript.

L257-259 I am lost. The effect of what is meant by its? 1 st sentence is unclear. 2nd sentence, I assume the frequency of mite infertility has been recognised by science, as a key element in understanding resistance. If it was long established, we would not have a problem. There exist many explanations, still exist nor were, I think, while I agree that the simplest is often the best.

L276 ..drone brood. Why is not so clear, any observations regarding hygienic behaviour towards drone cells?

Minor suggestions.

L9-10 First sentence of abstract can be deleted, is repeated in introduction.

L39 ..allowed for.., I suggest, ..frequently resulted in..

L104 adults, better known as: worker bees

L117 ..that were created.., I suggest ..that appear to be created..

L120 (35) un-superscript

L159 This, I assume this refers to the influx of mites, rather than the removal rate, however that is one sentence removed.

L182 ..as found, replace with ..similar to that found.. and add a quote at end of sentence.

L196 replace ..losses.. with ..loss rates..

L197 less or no greater than the losses(?) replace with ..below those .. and add .. in France (44)

L200 replacing ..single.. with ...complex... seems very appropriate

L213 replace .. is.. with ..may be..

L233 replace ..all NVR.. with all examined resistant..

L270 replace ..selected.. with bred

Referee: 3

Comments to the Author(s)

This is an interesting approach and a huge piece of work with the aim to use raw data from about 50 publications on Varroa resistance traits like VSH, SMR, recapping and others. However, I have some major concerns with the approach, the use of the available data and the conclusions:

1. The criteria for the natural Varroa resistant (NVR) colonies are not clearly defined. There are many data from colonies in South and Central America (Africanized honey bees) and Africa, mainly tropical regions with a huge population of wild-living honey bees. From the temperate regions there exist mostly small "populations", often not really stable over a long time period and/ or often not really defined in their structure and management.

At least it must be listed which populations are considered as NVR (and why).

2. A clear quality control of the data is lacking (or at least not described). Meanwhile there are so many papers on Varroa tolerance/ resistance that this is necessary. Furthermore it is problematic to merge data from publications of nearly 4 decades. For instance in Line 137 - 141 it is mentioned that there are two ways of measuring "infertility" and that only the definition "no viable female offspring" is used. According to the supplemental material there are definitively data included that used the parameter "no egg laying at all". This may not have huge impact on the difference between susceptible and NVR but demonstrates the general problem with "old" data, often not even properly described in the method section. The "bee book" was published exactly to overcome this problem, in the year 2013.

In addition, Fig. 2 obviously present data of single colonies? What are about published data that are average of several colonies without providing the single colony data? There are definitively such data in the literature of the suppl. mat. involved.

3. The data are biased because there are many data from a few authors (see suppl. material), many unpublished data.

4. I miss a critical discussion of the real result (which are presented more or less in Fig. 2). The discussion rather represents a solid review of the current discussion on the most important resistance traits.

5. The main question that are discussed in this manuscript is whether hygienic behaviour can explain also some other traits like SMR (suppressed mite reproduction). I agree that this is currently a "hot debate" (there exist already some reviews and opinion letters) and is important for the question which approach is suitable for the selection of resistant colonies. The authors obviously try to prove their assumption "that NVR colonies, regardless of location, have all developed the same strategy to control the mite". In this discussion, some studies are not included giving a differentiated view on the complex host-parasite relationship.

Some detailed comments:

1. L 57ff: This is overstated. In fact there are extremely few beekeeper stopping treatments (promoting natural selection) and most of them loose their colonies. Importantly, there are hardly good data from such activities.

2. L68: Why should this paper (a good and interesting paper mainly on recapping) be a "breakthrough"?

3. L108ff: It should be added for each parameter/ trait the number of colonies/ different locations which are used in the data set. It is too difficult to extract this from suppl. mat.

4. L131: Reproductive cycles between 2 and 3? There are other data from Fries (1996).

5. L136 (and L260ff): "increase in the number of infertile mites as they exhaust their limited supply of eggs". Here only two papers from 1984 + 1986 are cited. To my knowledge there exist no study confirming that the lack of oocytes in the Varroa female is a reason for infertility.

6. L148: Why is DWV removed from the equation?

7. L158: At least in temperate regions the critical infestation rates are quite lower. Are these statements supported by studies?

8. L177ff: These statements are not supported by literature!

9. L197: This generalized statement is only supported by one study on a small population in Southern France.

10. L1200ff: This statement must be explained and justified through the results.

11. L 267-276: These important statements are only justified by two papers from 2002 (one dealing with AFB)

12. L278ff: This statement is not supported by the data or the cited literature!

13. Some important papers/aspects are missing in the discussion, for instance:

Eynard et al., (2020) assuming that VSH and SMR are not linked!

Schöning et al. (2012): Removal of brood/ mites elicited by DWV symptoms.

Page et al. (2016, and others): "altruistic suicide" in *A. cerana* as an important trait of the original host.

Author's Response to Decision Letter for (RSPB-2020-2795.R0)

See Appendix A.

RSPB-2021-0519.R0

Review form: Reviewer 3

Recommendation

Accept as is

Scientific importance: Is the manuscript an original and important contribution to its field?

Good

General interest: Is the paper of sufficient general interest?

Excellent

Quality of the paper: Is the overall quality of the paper suitable?

Good

Is the length of the paper justified?

Yes

Should the paper be seen by a specialist statistical reviewer?

No

Do you have any concerns about statistical analyses in this paper? If so, please specify them explicitly in your report.

No

It is a condition of publication that authors make their supporting data, code and materials available - either as supplementary material or hosted in an external repository. Please rate, if applicable, the supporting data on the following criteria.

Is it accessible?

Yes

Is it clear?

Yes

Is it adequate?

N/A

Do you have any ethical concerns with this paper?

No

Comments to the Author

Congratulation to the authors, to my opinion the manuscript has significantly improved through the 3 reviews. The authors indeed tried to adress all my comments and criticisms. Most are satisfactory included in the manuscript, some points are sufficiently explained in the respnse to my first review.

I still do not completely agree with all statements and the general hypothesis on the extraordinary (or nearly xclusive) importance of hygienic behaviour for mite resistance all over the countries, however I consider this manuscript now a valuable and important contribution - using an innovative and elaborate approach for the meta analysis - to the current discussion on Varroa resistance and future strategies in selective breeding with a discussion that also includes the still open questions on host-parasite relationship.

Nearly unbelievable: I do not have detailed concerns on the reviewed manuscript.

Review form: Reviewer 2

Recommendation

Reject – article is scientifically unsound

Scientific importance: Is the manuscript an original and important contribution to its field?

Good

General interest: Is the paper of sufficient general interest?

Excellent

Quality of the paper: Is the overall quality of the paper suitable?

Acceptable

Is the length of the paper justified?

Yes

Should the paper be seen by a specialist statistical reviewer?

No

Do you have any concerns about statistical analyses in this paper? If so, please specify them explicitly in your report.

No

It is a condition of publication that authors make their supporting data, code and materials available - either as supplementary material or hosted in an external repository. Please rate, if applicable, the supporting data on the following criteria.

Is it accessible?

N/A

Is it clear?

N/A

Is it adequate?

N/A

Do you have any ethical concerns with this paper?

No

Comments to the Author

The authors present their results of a metanalysis of 58 papers on the topic of honey bee resistance to the ectoparasitic mite *Varroa*. This approach provide data across continent all over the world and all the spectrum of beekeeping conditions including climate and environment. It turn out that all the resistant populations have associated the following 3 traits: recapping, brood removal and reduced mite reproduction. It has implication in the understanding of honey bee resistance to their parasites to eventually reduced treatments or even keep bees without the need of any treatments while maintaining performances.

This is a timely topic and would be a nice contribution to the fields. The paper is well written in a synthetic straight to the point way what in turn ease the read flow.

I have however several concerns

General comments

First of all I have concerns about the structure of the manuscript.

I was expecting a research article. As said, it is well written, in a nice English, in a clear logic.

While reading I quickly realized I was more and more confused and I had more and more difficulties to understand. At the end, I have the feeling maybe it would be better to make a review? Or in a format where you can do in the same time Results in Discussion? As it stand, to my opinion it is not a structure of a Research Paper. I know there is no clear definition about a definition if a literature metanalysis better fit as Research article or Review. But nothing stand in the way of doing a metanalysis in a Review. Here you quote more than 160 papers...

Also you results part is partially already a discussion. It is not only about presenting the results of the framework you build but also put in perspective with the literature. SO sure you research is using papers to extract information out of it, so when you comment your results, in a way you talk about the different paper you used and you quote them. But in the results part you don't quote at all the 58 paper you used for the frame work, but 49 references from literature. It sounds a lot to me for a results section. I would expect in a Research paper you present only of the results per se, and then go to the discussion and put in perspective and quote a lot of refs there.

In addition to me, your testable hypothesis is not clear. Actually, you looked resistant population in literature and try to found out common resistant trait. However, the way you present your aim sounds that resistance is assumed to be fixed and established and that out of the literature it will demonstrate that the evolution fixed in population resistance traits. At the end, I see an aim but not a hypothesis, so that also there is a discrepancy with the title of the paper.

As I mentioned earlier this is hot and timely topic. A lot was published lately. I have the feeling that many references are missing in your reference list. Starting with nice reviews hat has been recently published: Le Conte, Buchler et al in *Insects*, Mondet et al., Traynor et al, Guichard et al. Actually the last one is a review that study in detail the resistance topic and that I would have expect to be mentioned at least in the discussion. Indeed there is plenty to discuss. Just one example: surviving colony or population does not necessary imply that the colony is resistant or tolerant to *Varroa*. It could also be due to environmental reasons. When you move a "resistant" colony in a new place, suddenly the colony is not anymore resistant at all...

Additionally, you maybe might develop more the methods. It is a bit short and confusing. You might want make a figure or table to make it easier for the reader. For example you start with "Using all the current available data"... But at this point I have no idea about which data you are talking about.

Then since you take paper all over the world, this indeed mean Different type of bees, Different *Varroa* Haplotypes, Different populations, different protocols to measure the traits (without talking about accuracy and reliability of the measurements), so in one word different conditions. In addition, what about environmental effects? This make impossible to compare the data. That says I know you are not comparing. More putting together, aggregating data. Therefore, I would expect you validate the way you worked or explain why you believe it worked to put all of this together and might be analysed and interpreted...

Finally, in your discussion, about the topic of at least 7 countries and Hygienic behavior it is not very clear to me. Which seven countries are you talking? Could you clearly mention the refs?

I do not see how you results can lead to demonstrate what you say in the discussions....

There is 30 years of research on Hygienic behavior and so far no clear link direct and indirect with *Varroa* resistance in the field. There is no selection program using Hygienic behavior that succeeded to improve resistance. And there is no clear link on Hygienic Behavior and survival of hives (rather negative). There is no clear correlation between Hygienic behavior and *Varroa*

resistance in the field. (In Africa, Kenya for example, or a recent review of Spivak and Danka, in *Apidologie*, 2021, or Leclercq et al., 2017 and 2018 or Guichard et al., 2020)

If seen over continents, results are more about recapping results. Alternatively, maybe I did not understand what you meant with Hygienic Behavior. So in this case please add a clear definition. In all resistant population, High Hygienic Behavior has not been systematically demonstrated.

Detailed comments

Maybe you put too much emphasis on DWV in the abstract and Introduction for then not really talk about it in results (a bit in Honey bee behavior, and Colony level effects; but without so much details or dedicated section) and then in Discussion (but not so detailed)...

Sometime you say framework and sometime metanalysis- Maybe I misunderstood...maybe stick with one? I would say metanalysis?

I would suggest to well establish definitions. For example: Until you say in method resistance is surviving more than 5 years, I thought it meant reducing the number of parasite in the population.

I see that you avoid common traits name such VSH or SMR or more recently MNR. Which is actually a good idea. Because sometime people use the word of VSH for SMR or vice versa etc. And there is still not really a standard terminology on the field...So indeed it is a good idea to be more general and saying like you do brood removal or infertility, or reduced mite reproduction. I would say they is room for improvement: maybe you could try to better clarify your terminology/nomenclature and make it clear in the text all your definitions and how the reader should understand it.

L.39: what is "this". What does it refer to?. Not clear. Specify.

L.45: interesting and important point. I would expect more refs here.

L48: Maybe this is not that easy as it sound. Here I disagree. What about haplotype? Here in Fernando de Noronha we are talking about Japanese Haplotype. And probably this might be the only population known with Varroa Tolerance.....

L58: I do not fully get why you distinguish Europe from Uk. Geographically speaking this is the same region, and might probably all together be described as Europe. Why not say two regions (one side of the Atlantic, and the other Side with USA).

L61. Which one are you talking about? Not clear here. I don't uderstanbd what are those 7 populations....

L.68: based on the introduction it is not clear that it arised. It does sound like a fact. But is that really sure they are truly resistant? If so, when moved in a new locations, should those populations still not be resistant? In it is not necessary the cases....

Result section: Could you precise: was the 3 traits always in all the resistant populations? Or was that one of them? Which one? This is very important, because so far I do not know any "universal" traits that work in any populations.

L240: I am unsure about the word "demonstrate". Maybe reformulate with something like suggests. Or if you want underline then maybe " shows".

L274: How do you know the Freeze killed is the most commonly used method for hygienic behavior? And where? In Europe, US, where else or the World?

L275: this is maybe a style issue, and each one his own style. But I would not expect to see “don’t” in a scientific publication but rather “do not” in full word

L307: the paper quoted here start to be a little bit old. Is there anything newest as well? And it deals mainly with VSH. So do they really looked at removing and recapping? Did they see it once? Or in only one population? Or a lot? I mean maybe it is hard to generalize, extrapolate or effent say this sentence based on one paper, or?

L 316. Why is jacobsoni between brackets?

L337: Like line 275, I would expect full wording instead of “won’t”

L338: well I am not sure is it possible to say “same solution”...Or at least on my opinion your data are not sufficient to say that. You might want maybe reformulate...

References

Look good! I did not see any problems or typos...well done

Supplementary data:

It is well prepared and presented.

Well done.

Thank you for your work

Decision letter (RSPB-2021-0519.R0)

29-Apr-2021

I am writing to inform you that this version of your manuscript RSPB-2021-0519 entitled "Parallel Evolution of Varroa Resistance in honey bees; a common mechanism across continents" has, in its current form, been rejected for publication in Proceedings B.

This action has been taken on the advice of referees, who have recommended that substantial revisions are necessary. With this in mind we would be happy to consider a resubmission, provided the comments of the referees are fully addressed. However please note that this is not a provisional acceptance.

Please find below the comments made by the referees, not including confidential reports to the Editor, which I hope you will find useful.

- 1) A ‘response to referees’ document including details of how you have responded to the comments, and the adjustments you have made.
- 2) A clean copy of the manuscript and one with 'tracked changes' indicating your 'response to referees' comments document.

- 3) Line numbers in your main document.
- 4) Please read our data sharing policies to ensure that you meet our requirements <https://royalsociety.org/journals/authors/author-guidelines/#data>.

Sincerely,
Dr Locke Rowe
mailto: proceedingsb@royalsociety.org

Associate Editor
Comments to Author:

Your manuscript has been seen by one of the original reviewers, and a 4th reviewer. As you can see, the latter has raised a number of issues, both major and minor.

I have taken this most recent review, and the previous reviews, and gone back to your manuscript and considered the key issues. From this, a number of key points arise:

1) you state that you have conducted a meta-analysis. However, there are very clear methods for such an analysis and you have not followed them (see <http://www.prisma-statement.org> and <https://doi.org/10.1038/s41559-020-01295-x>). In particular, you do not report how you searched the literature (search terms, engine, date, papers collected and rejected at each stage, etc.), and your statistical methodology does not hold up from a meta-analysis perspective (you do not account for sample sizes, use Hedges' D to look at effect sizes, etc. etc.). You either need to report all of this information and do correct statistical analyses, or avoid using the term 'meta-analysis'. Either way, you still need to report how you searched for and screened the papers you base your analyses and model on - without this your study is not repeatable, and if it's not repeatable then it is not good science

2) your results come in two forms (i) the statistical analyses of the datasets you have gathered, and (ii) a model putting all of these together which acts as a hypothesis to explain resistance to Varroa. At the moment, the way these are presented is confusing. You put the model first as a figure (with no preceding text) and then your results section bounces back and forth between the actual results and the model that you put together from them. This is not helpful - indeed, is actually confusing - to a reader. I would suggest that this section needs complete restructuring, to present the analyses of published data first, and then the construction of the model, with explicit acknowledgement that the model is a hypothesis

3) the justification for the structure of the model is lacking. Given that this is the key outcome of your study, the justification needs to be clear and explicit. Why is this the only, or the best way, to put the data together and interpret them? You're basically asking your reader to take all of this on faith. As an example, lines 202-203 make a huge factual claim, but it is actually an interpretation of the data not a fact.

4) significant sections of the Results should be in the Discussion (e.g., lines 615-630)

5) the Discussion starts far too strongly - the sentence in lines 1308-1308 is a much more measured statement of what you have found. Fundamentally, you have analysed data, shown differences between resistant and susceptible populations, and created a model framework from this to explain resistance that now needs to be explicitly tested. There is real value to this, but this is not how you present your work.

In addition to these issues, the points made by the 4th reviewer (some of which overlap with my comments above) need addressing. I hope you find this useful as you revise your manuscript.

Reviewer(s)' Comments to Author:

Referee: 3

Comments to the Author(s).

Congratulation to the authors, to my opinion the manuscript has significantly improved through the 3 reviews. The authors indeed tried to adress all my comments and criticisms. Most are satisfactory included in the manuscript, some points are sufficiantly explained in the respnse to my first review.

I still do not completely agree with all statements and the general hypothesis on the extraordinary (or nearly xclusive) importance of hygienic behaviour for mite resistance all over the countries, however I consider this manuscript now a valuable and important contribution - using an innovative and elaborate approach for the meta analysis - to the current discussion on Varroa resistance and future strategies in selective breeding with a discussion that also includes the still open questions on host-parasite relationship.

Nearly unbelievable: I do not have detailed concerns on the reviesd manuscript.

Referee: 4

Comments to the Author(s).

The authors present their results of a metananalysis of 58 papers on the topic of honey bee resistance to the ectoparasitic mite Varroa. This approach provide data across continent all over the world and all the spectrum of beekeeping conditions including climate and environment. It turn out that all the resistant populations have associated the following 3 traits: recapping, brood removal and reduced mite reproduction. It has implication in the understanding of honey bee resistance to their parasites to eventually reduced treatments or even keep bees without the need of any treatments while maintaining performances.

This is a timely topic and would be a nice contribution to the fields. The paper is well written in a synthetic straight to the point way what in turn ease the read flow.

I have however several concerns

General comments

First of all I have concerns about the structure of the manuscript.

I was expecting a research article. As said, it is well written, in a nice English, in a clear logic.

While reading I quickly realized I was more and more confused and I had more and more difficulties to understand. At the end, I have the feeling maybe it would be better to make a review? Or in a format where you can do in the same time Results in Discussion? As it stand, to my opinion it is not a structure of a Research Paper. I know there is no clear definition about a definition if a literature metanalysis better fit as Research article or Review. But nothing stand in the way of doing a metanalysis in a Review. Here you quote more than 160 papers...

Also you results part is partially already a discussion. It is not only about presenting the results of the framework you build but also put in perspective with the literature.SO sure you research is using papers to extract information out of it, so when you comment your results, in a way you talk about the different paper you used and you quote them. But in the results part you don't quote at all the 58 paper you used for the frame work, but 49 references from literature. It sounds a lot to me for a results section.I would expect in a Research paper you present only of the results per se, and then go to the discussion and put in perspective and quote a lot of refs there.

In addition to me, your testable hypothesis is not clear. Actually, you looked resistant population in literaute and try to found out common resistant trait. However, the way you present your aim

sounds that resistance is assumed to be fixed and established and that out of the literature it will demonstrate that the evolution fixed in population resistance traits. At the end, I see an aim but not a hypothesis, so that also there is a discrepancy with the title of the paper.

As I mentioned earlier this is hot and timely topic. A lot was published lately. I have the feeling that many references are missing in your reference list. Starting with nice reviews that has been recently published: Le Conte, Buchler et al in *Insects*, Mondet et al., Traynor et al, Guichard et al. Actually the last one is a review that study in detail the resistance topic and that I would have expect to be mentioned at least in the discussion. Indeed there is plenty to discuss. Just one example: surviving colony or population does not necessary imply that the colony is resistant or tolerant to *Varroa*. It could also be due to environmental reasons. When you move a “resistant” colony in a new place, suddenly the colony is not anymore resistant at all...

Additionally, you maybe might develop more the methods. It is a bit short and confusing. You might want make a figure or table to make it easier for the reader. For example you start with “Using all the current available data” ...But at this point I have no idea about which data you are talking about.

Then since you take paper all over the world, this indeed mean Different type of bees, Different *Varroa* Haplotypes, Different populations, different protocols to measure the traits (without talking about accuracy and reliability of the measurements), so in one word different conditions. In addition, what about environmental effects? This make impossible to compare the data. That says I know you are not comparing. More putting together, aggregating data. Therefore, I would expect you validate the way you worked or explain why you believe it worked to put all of this together and might be analysed and interpreted...

Finally, in your discussion, about the topic of at least 7 countries and Hygienic behavior it is not very clear to me. Which seven countries are you talking? Could you clearly mention the refs? I do not see how you results can lead to demonstrate what you say in the discussions.... There is 30 years of research on Hygienic behavior and so far no clear link direct and indirect with *Varroa* resistance in the field. There is no selection program using Hygienic behavior that succeeded to improve resistance. And there is no clear link on Hygienic Behavior and survival of hives (rather negative). There is no clear correlation between Hygienic behavior and *Varroa* resistance in the field. (In Africa, Kenya for example, or a recent review of Spivak and Danka, in *Apidologie*, 2021, or Leclercq et al., 2017 and 2018 or Guichard et al., 2020) If seen over continents, results are more about recapping results. Alternatively, maybe I did not understand what you meant with Hygienic Behavior. So in this case please add a clear definition. In all resistant population, High Hygienic Behavior has not been systematically demonstrated.

Detailed comments

Maybe you put too much emphasis on DWV in the abstract and Introduction for then not really talk about it in results (a bit in Honey bee behavior, and Colony level effects; but without so much details or dedicated section) and then in Discussion (but not so detailed)...

Sometime you say framework and sometime metanalysis- Maybe I misunderstood...maybe stick with one? I would say metanalysis?

I would suggest to well establish definitions. For example: Until you say in method resistance is surviving more than 5 years, I thought it meant reducing the number of parasite in the population.

I see that you avoid common traits name such VSH or SMR or more recently MNR. Which is actually a good idea. Because sometime people use the word of VSH for SMR or vice versa etc. And there is still not really a standard terminology on the field...So indeed it is a good idea to be more general and saying like you do brood removal or infertility, or reduced mite reproduction. I would say they is room for improvement: maybe you could try to better clarify your

terminology/nomenclature and make it clear in the text all your definitions and how the reader should understand it.

L.39: what is "this". What does it refer to?. Not clear. Specify.

L.45: interesting and important point. I would expect more refs here.

L48: Maybe this is not that easy as it sound. Here I disagree. What about haplotype? Here in Fernando de Noronha we are talking about Japanese Haplotype. And probably this might be the only population known with Varroa Tolerance.....

L58: I do not fully get why you distinguish Europe from Uk. Geographically speaking this is the same region, and might probably all together be described as Europe. Why not say two regions (one side of the Atlantic, and the other Side with USA).

L61. Which one are you talking about? Not clear here. I don't udnerstanbd what are those 7 populations....

L.68: based on the introduction it is not clear that it arised. It does sound like a fact. But is that really sure they are truly resistant? If so, when moved in a new locations, should those populations still not be resistant? In it is not necessary the cases....

Result section: Could you precise: was the 3 traits always in all the resistant populations? Or was that one of them? Which one? This is very important, because so far I do not know any "universal" traits that work in any populations.

L240: I am unsure about the word "demonstrate". Maybe reformulate with something like suggests. Or if you want underline then maybe "shows".

L274: How do you know the Freeze killed is the most commonly used method for hygienic behavior? And where? In Europe, US, where else or the World?

L275: this is maybe a style issue, and each one his own style. But I would not expect to see "don't" in a scientific publication but rather "do not" in full word

L307: the paper quoted here start to be a little bit old. Is there anything newest as well? And it deals mainly with VSH. So do they really looked at removing and recapping? Did they see it once? Or in only one population? Or a lot? I mean maybe it is hard to generalize, extrapolate or effent say this sentence based on one paper, or?

L 316. Why is jacobsoni between brackets?

L337: Like line 275, I would expect full wording instead of "won't"

L338: well I am not sure is it possible to say "same solution"...Or at least on my opinion your data are not sufficient to say that. You might want maybe reformulate...

References

Look good! I did not see any problems or typos...well done

Supplementary data:

It is well prepared and presented.

Well done.

Thank you for your work

Author's Response to Decision Letter for (RSPB-2021-0519.R0)

See Appendices B & C.

RSPB-2021-1375.R0

Review form: Reviewer 4

Recommendation

Accept with minor revision (please list in comments)

Scientific importance: Is the manuscript an original and important contribution to its field?

Good

General interest: Is the paper of sufficient general interest?

Good

Quality of the paper: Is the overall quality of the paper suitable?

Good

Is the length of the paper justified?

Yes

Should the paper be seen by a specialist statistical reviewer?

No

Do you have any concerns about statistical analyses in this paper? If so, please specify them explicitly in your report.

No

It is a condition of publication that authors make their supporting data, code and materials available - either as supplementary material or hosted in an external repository. Please rate, if applicable, the supporting data on the following criteria.

Is it accessible?

Yes

Is it clear?

Yes

Is it adequate?

Yes

Do you have any ethical concerns with this paper?

No

Comments to the Author

The authors take advantages of published data of 60 publications to better understand Varroa resistance evolution in *Apis mellifera*. They compare susceptible colonies and resistant, focusing on 3 phenotypes Brood removal, recapping and mite infertility. Based on the results they constructed a Framework to link the traits and deduce mechanisms. The knowledge pave the way to enhance selection efficiency in breeding programs towards Varroa resistant colonies.

This an interesting work, that scientifically sounds.
Thank you to the authors for their work.

General comments

This is a timely topics that indeed need to study in deep to offer solution in the field in a near future. The results here stimulate thinking, and the ideas discussed here bring fuel to the debate. I think the paper is clear, correctly build and straight forward. Sure there is convergence in many place in the world, but still on my opinion, it stay at this stage in this paper correlative results, and not prove of mechanisms. So that the concepts and idea brought here are nice, but still need to be proven in field.

Detail comments:

L14: I do see the S at the end of populations. Still this sound that many population around the globe are surviving, the majority. But still probably, at least in the managed colony with western honey bee in occidental country, the vast majority need treatments. Only the minority is surviving....

L 19: I would suggest to remove " for the first time". To write this does not make your MS any better.

L24: Not sure what is Natural Varroa Resistance here. It come then only one time in the text about Fernando de Noronha, but that's it. Not definition. But you define in the other hand resistant population L 92. Is that the same? I believe so...Here definition is important since the reader may think that resistance bee are colonies that have load Varroa load and survive, in opposition of Tolerance with high varroa load.

L 55: probably I would expect to fully write the word and not "don't".

L.90: Actually why directly focuses on this 3 specific traits. Not very clear to me. I would expect that you explore all possible Varroa phenotype, only then you see the 3 best, or the one that are the most important, or in common in all populations....

L105-106: Probably fine to leave it here, but it could be also place in the results part.

L135 - 136_ Pre or Per?

Results

Throughout the results part and also later in the article, you talk about figure 1 . But then it is figure 8 in the Line 216

Discussion:

Globally it is a nice discussion.

The beginning maybe does not fit very well with the goal of l 79-83. I was expecting something like all population have in common three traits and you will discussed that here. But I had the feeling reading the first paragraph that you would like to show how Varroa resistance arise in those population. And you emphasize the framework. Make sense to discuss about it. I might have misunderstood, but my understanding is that it is not the discussion of the results per se, but more your vision. Sure it is an attempt. But maybe I would have expect the discussion about that and speculation later in the discussion part of the manuscript.

Decision letter (RSPB-2021-1375.R0)

08-Jul-2021

Dear Miss Grindrod

I am pleased to inform you that your manuscript RSPB-2021-1375 entitled "Parallel Evolution of Varroa Resistance in Honey Bees; a common mechanism across continents?" has been accepted for publication in Proceedings B.

The referee(s) have recommended publication, but also suggest some minor revisions to your manuscript. Therefore, I invite you to respond to the referee(s)' comments and revise your manuscript. Because the schedule for publication is very tight, it is a condition of publication that you submit the revised version of your manuscript within 7 days. If you do not think you will be able to meet this date please let us know.

Sincerely,
Dr Locke Rowe
mailto: proceedingsb@royalsociety.org

Associate Editor
Board Member
Comments to Author:

Thank you for the major revisions you have made to your manuscript, which is much improved as a result. The reviewer makes a couple of recommendations for small textual changes that you need to make. I do not expect you to rewrite or restructure your discussion as suggested, as it does work as is, but if you believe that responding to the discussion comment from the reviewer will improve this section, please do follow it. In addition, please check your manuscript throughout for spelling errors. Thanks for engaging with the review process in such a positive way.

Reviewer(s)' Comments to Author:

Referee: 4

Comments to the Author(s).

The authors take advantages of published data of 60 publications to better understand Varroa resistance evolution in *Apis mellifera*. They compare susceptible colonies and resistant, focusing on 3 phenotypes Brood removal, recapping and mite infertility. Based on the results they constructed a Framework to link the traits and deduce mechanisms. The knowledge pave the way to enhance selection efficiency in breeding programs towards Varroa resistant colonies.

This an interesting work, that scientifically sounds.
Thank you to the authors for their work.

General comments

This is a timely topics that indeed need to study in deep to offer solution in the field in a near future. The results here stimulate thinking, and the ideas discussed here bring fuel to the debate. I think the paper is clear, correctly build and straight forward. Sure there is convergence in many place in the world, but still on my opinion, it stay at this stage in this paper correlative results, and not prove of mechanisms. So that the concepts and idea brought here are nice, but still need to be proven in field.

Detail comments:

L14: I do see the S at the end of populations. Still this sound that many population around the globe are surviving, the majority. But still probably, at least in the managed colony with western honey bee in occidental country, the vast majority need treatments. Only the minority is surviving....

L 19: I would suggest to remove " for the first time". To write this does not make your MS any better.

L24: Not sure what is Natural Varroa Resistance here. It come then only one time in the text about Fernando de Noronha, but that's it. Not definition. But you define in the other hand resistant population L 92. Is that the same? I believe so...Here definition is important since the reader may think that resistance bee are colonies that have load Varroa load and survive, in opposition of Tolerance with high varroa load.

L 55: probably I would expect to fully write the word and not "don't".

L.90: Actually why directly focuses on this 3 specific traits. Not very clear to me. I would expect that you explore all possible Varroa phenotype, only then you see the 3 best, or the one that are the most important, or in common in all populations....

L105-106: Probably fine to leave it here, but it could be also place in the results part.

L135 - 136_ Pre or Per?

Results

Throughout the results part and also later in the article, you talk about figure 1 . But then it is figure 8 in the Line 216

Discussion:

Globally it is a nice discussion.

The beginning maybe does not fit very well with the goal of l 79-83. I was expecting something like all population have in common three traits and you will discussed that here. But I had the feeling reading the first paragraph that you would like to show how Varroa resistance arise in those population. And you emphasize the framework. Make sense to discuss about it. I might have misunderstood, but my understanding is that it is not the discussion of the results per se, but more your vision. Sure it is an attempt. But maybe I would have expect the discussion about that and speculation later in the discussion part of the manuscript.

Author's Response to Decision Letter for (RSPB-2021-1375.R0)

See Appendix D.

Decision letter (RSPB-2021-1375.R1)

12-Jul-2021

Dear Miss Grindrod

I am pleased to inform you that your manuscript entitled "Parallel Evolution of Varroa Resistance in Honey Bees; a common mechanism across continents?" has been accepted for publication in Proceedings B.

Data Accessibility section

Open Access

Paper charges

Sincerely,
Proceedings B
<mailto:proceedingsb@royalsociety.org>

Appendix A

Referee: 1

Comments to the Author(s)

This is an excellent synthesis of prior research and new findings on honey bee mechanisms of resistance to *Varroa destructor* mites. Figure 1 and especially Fig 2 are brilliant. The take-home message for me is that it doesn't take a lot of one kind of behavior (e.g., average of 46% removal of mite-infested brood, 61% recapping) to lead to reduced mite fertility, reduced mite load, viral load and colony survival. The selection for mites preferring drone brood over worker brood due to the bees surveillance and disruption of mite-infested worker brood (and no surveillance or disruption of drone brood) makes a lot of sense, but requires more future study.

My only bone to pick is about the word "learning" (lines 101, 104, 200), and the assumption that uncapping cells is a rare trait (lines 120, 203). Honey bees uncap diseased brood: they uncap (but don't always remove) some portion of brood infected with chalkbrood, AFB, virus infected brood, and they uncap cells that have wax moth larvae tunneling under the wax. Uncapping and recapping are traits that predate *Varroa* in *Apis mellifera*. If a population of colonies has little or no disease, it might appear that uncapping/ recapping is rare. But from my experience, the most unhygienic of colonies uncap diseased brood – the problem occurs if they try to remove the brood when infectious, or even if they leave the infected brood in place and other bees come into contact with it, the uncapping/ removal increases the risk of pathogen transmission. My point is that I don't think uncapping and recapping are learned behaviours; they are part of the honey bee behavioural repertoire, but genetically some colonies are more prone to detect and uncap more quickly than other colonies. That said, I do think some learning may occur within a colony that is infected with a pathogen or infested with mites. Once cells have been opened, other bees may become "primed" or sensitized to the olfactory stimulus, but to my knowledge this has not been tested.

Amended – we agree, recapping predates *Varroa* and this has been now mentioned in the discussion L245-246

Very minor points:

Line 151: should say 40% of the infested pupae? Amended – so now reads 'more than 40% of infested pupae'

Line 213: tasks are performed by bees (plural) of different ages... Amended

Nice work.

Referee: 2

Comments to the Author(s)

The manuscript is based on the compilation of the data, I think, from all available literature published over the past several decades. As such, the main novelty is in the meta-analysis of the resulting massive dataset. It creates new learnings from previous publications, which is applaudable.

I have a few observations, that I think could improve the manuscript. I am always disturbed by the use of not generally accepted acronyms. In the current publication, NVR, FKB and AHB are used, the latter two for Freeze killed brood respectively Africanized honey bees only occurs few times, and really does not require abbreviation. NVR stands in for naturally varroa resistant. I have two issues with this. I can't be expected to learn a set of new acronyms for each paper I read, and it really reduces the readability for the casual reader, that just wants to gloss over the paper in 10 weeks or 20 years time, they may just put the paper down, as being unreadable. More important than my discomfort, in most parts of the paper, naturally varroa resistant bees are discussed as opposed to susceptible bees. So the logic question is, are susceptible bees unnatural? Obviously not. They are susceptible whereas the other bees are resistant. Only in the second large paragraph, L267 to L276 are bees discussed, that have been bred for high level of hygienic behaviour, but does that really qualify as unnatural?

We agree with this good comment and have changed NVR to 'resistant', FKB has been changed to 'freeze killed brood', and AHB has been changed to 'Africanised honey bees' as suggested as it makes it easier to read.

Figure 1 and figure 2 are having a very similar message, I think figure 2 is sufficient, maybe it could be improved by adding blue and orange borders around the figures, as in figure 1.

- and excellent suggestion so removed Fig. 1 and enhanced Fig. 2 as suggested.

L113-4 instead the comma and the next sentence, I suggest you write: Europe. Comparisons to susceptible colonies where Africanized and African bees occur is impossible, as these haven't be found in time to be assessed.

Amended we changed it to: Unfortunately, comparisons to susceptible colonies originating from where Africanized and African bees occur were not possible, as such populations haven't been found. (L125-127)

Paragraph 163-171. The reduction of DWV that results from resistant bees, is here only described as a general lower titre. The prevalence of DWV sick bees probably is reduced too, inside the colony, which has been shown relevant for survival (Dainat B, Neumann P. Clinical signs of deformed wing virus infection are predictive markers for honey bee colony losses. *Journal of invertebrate pathology*. 2013 Mar 1;112(3):278-80 and Francis RM, Nielsen SL, Kryger P. Varroa-virus interaction in collapsing honey bee colonies. *PLoS one*. 2013 Mar 19;8(3):e57540.).

We have now improved this section (L180-185) by including the two suggested papers and a very recent paper showing brood cannibalism can spread DWV allowing prevalence to remain high but since transmission via injection is more ineffective than feeding titres decrease.

This brings up another issue, that the authors could reflect more on. The graphs in figure 2 show a considerable level of variation, that by far exceeds the differences in means between susceptible and resistant bees. Partly this may be due to biological variation, however, as was recently shown, it is very hard to accurately measure hygienic behaviour (Mondet F, Parejo M, Meixner MD, Costa C, Kryger P, Andonov S, Servin B, Basso B, Bieńkowska M, Bigio G, Căuia E. Evaluation of Suppressed Mite Reproduction (SMR) Reveals Potential for Varroa Resistance in European Honey Bees (*Apis mellifera* L.). *Insects*. 2020 Sep;11(9):595.). I think the problem of measurements is an important discussion, and given the level of variability in the published dataset, that could deserve a paragraph in the results part of the manuscript.

We have now inserted a new paragraph (L220-233) entitled 'Variability of data' as suggested and included the Mondet reference along with several others.

L257-259 I am lost. The effect of what is meant by its? 1 st sentence is unclear. 2nd sentence, I assume the frequency of mite infertility has been recognised by science, as a key element in understanding resistance. If it was long established, we would not have a problem. There exist many explanations, still exist nor were, I think, while I agree that the simplest is often the best.

Thanks for pointing this out, we have completed reworded this paragraph and introduced more detail and references as requested by reviewer 3. (L285-295)

L276 ..drone brood. Why is not so clear, any observations regarding hygienic behaviour towards drone cells?

There are limited observations concerning drone brood, it appears that both in *A. cerana* and *A. mellifera* infested drone cells are typically neglected, i.e. no removal nor recapped.

(Mentioned at L302- 303) We have re-written this section as requested by reviewer 3 and discussed the role of drone brood in more detail (Paragraph: L297 to L310).

Minor suggestions.

L9-10 First sentence of abstract can be deleted, is repeated in introduction. **It has been deleted**

L39 ..allowed for.., I suggest, ..frequently resulted in.. **Amended**

L104 adults, better known as: worker bees **Amended**

L117 ..that were created.., I suggest ..that appear to be created.. **Amended**

L120 (35) un-superscript **Amended**

L159 This, I assume this refers to the influx of mites, rather than the removal rate, however that is one sentence removed.

It was the removal rate but if one assumes that bees remove all of the infested cells then these figures could be applied to mite influx, section reworded – L173-176

L182 ..as found, replace with ..similar to that found.. and add a quote at end of sentence. **Amended**

L196 replace ..losses.. with ..loss rates.. **Amended**

L197 less or no greater than the losses(?) replace with ..below those .. and add .. in France (44) **Amended**

L200 replacing ..single.. with ...complex... seems very appropriate **Amended**

L213 replace .. is.. with ..may be.. **Amended**

L233 replace ..all NVR.. with all examined resistant.. **Amended**

L270 replace ..selected.. with bred **Amended**

Referee: 3

Comments to the Author(s)

This is an interesting approach and a huge piece of work with the aim to use raw data from about 50 publications on *Varroa* resistance traits like VSH, SMR, recapping and others. However, I have some major concerns with the approach, the use of the available data and the conclusions:

1. The criteria for the natural *Varroa* resistant (NVR) colonies are not clearly defined. There are many data from colonies in South and Central America (Africanized honey bees) and Africa, mainly tropical regions with a huge population of wild-living honey bees. From the temperate regions there exist mostly small "populations", often not really stable over a long time period and/ or often not really defined in their structure and management.

At least it must be listed which populations are considered as NVR (and why).

Amended- we have replaced the term NVR for 'resistant' as it was confusing the situation. We now define resistant bees as those that have *Varroa* and have survived for at least 5 years without any sort of treatment for mites (see L75) within the methods, although many populations in and UK and Europe have not been treated for >10 years. We had collected data from resistant traits from across 3 South & Central American, 3 African and 5 European countries. As there are no susceptible colonies in South/Central American, and Africa only resistant data is available, but we have data from resistant colonies from 5 European countries. As requested we have now provided the country of origin for all resistant and susceptible colonies in the supplementary data.

2. A clear quality control of the data is lacking (or at least not described). Meanwhile there are so many papers on *Varroa* tolerance/ resistance that this is necessary.

Our quality control is now included within the methods:

L79 – 86 reads: In order to standardise the data studies had to use natural comb and any infested cells contained a single live mother-mite. For recapping, brood removal and mite non-reproduction data the minimum sample size per colony was 50 cells or mites. Where possible single colony data was extracted from the studies. All recapping data (n = 140) came from single colonies; for brood removal seven of the 83 data points are colony averages; for mite infertility 75 of the 102 data points are colony averages (see supplementary data).

Furthermore it is problematic to merge data from publications of nearly 4 decades. For instance in Line 137 - 141 it is mentioned that there are two way of measuring "infertility" and that only the definition "no viable female offspring" is used.

We have now clarified this in the text since as originally mentioned we used the less stringent definition (no viable offspring) as this also encompassed papers that used the more severe definition (no offspring at all). L150-158

According to the suppl. material there are definitively data included that used the parameter "no egg laying at all". This may not have huge impact on the difference between susceptible and NVR but demonstrate the general problem with "old" data, often not even properly described in the method section. The "bee book" was published exactly to overcome this problem, in the year 2013.

The methodology of all old studies used was carefully checked to make sure it was acceptable and the reviewer is correct that it does not have a huge impact on the results.

In addition, Fig. 2 obviously present data of single colonies? What are about published data that are average of several colonies without providing the single colony data? There are definitively such data in the literature of the suppl. mat. involved.

We have now made it clear both in the methods (L83-86) and supplementary data the number of single and multiple colonies that provided the data. Actually 75% of the 325 data points came from single colonies.

3. The data are biased because there are many data from a few authors (see suppl. material), many unpublished data.

We have included data from all studies, so long as they met the minimal standards we set out in the methods. The data for the infested-brood removal was gathered by 6 different main authors and contributed equally; for recapping the data comes from 2 groups of authors this reflects the recent importance of this trait since prior to the Oddie et al. 2018 paper recapping was mentioned but data was never collected. Despite this these two teams have collected data from 7 countries using the same method, and finally the mite-infertility data comes from over 16 different research teams.

The only unpublished data comes from a single author Dr Luis Medina 2019 from an ongoing study at his institution, he has published several studies on Varroa and the inclusion of this unpublsh dataset allows the direct comparison within the same population of decreasing varroa infestation rates, the data which will eventually be published. For clarification we have now added a footnote to the table S4 explaining where, and by whom the unpublished data was collected and why it was included.

4. I miss a critical discussion of the real result (which are presented more or less in Fig. 2). The discussion rather represents a solid review of the current discussion on the most important resistance traits.

We have now expanded the discussion taking onboard all the advice from the three reviewers. Also we have now indicated how this study meta-analysis differs from and advances knowledge from recent reviews on the subject published after this manuscript was submitted.

5. The main question that are discussed in this manuscript is whether hygienic behaviour can explain also some other traits like SMR (suppressed mite reproduction). I agree that this is currently a "hot debate" (there exist already some reviews and opinion letters) and is important for the question which approach is suitable for the selection of resistant colonies. The authors obviously try to prove their assumption "that NVR colonies, regardless of location, have all developed the same strategy to control the mite". In this discussion, some studies are not included giving a differentiated view on the complex host-parasite relationship.

In the expanded discussion all the reviewer suggested papers are included, plus some others to discuss more broadly the complex host-parasite relationship and highlight areas where more data are needed.

Some detailed comments:

1. L 57ff: This is overstated. In fact there are extremely few beekeeper stopping treatments (promoting natural selection) and most of them lose their colonies. Importantly, there are hardly good data from such activities.

We agree that there is a lack of data thus the authors have been conducting a survey among the British beekeepers, to date we have had over 2500 replies and these preliminary results indicate 10% have not been treating for Varroa in the past six years, and 18% having not treated in the past 1 to 5 year. However, currently this is unpublished data and we agree that in most northern hemisphere countries the number is likely to be much lower than the UK. Therefore we now say: '**very** small but increasing number'.

2. L68: Why should this paper (a good and interesting paper mainly on recapping) be a "breakthrough"?

Boecking (1990) first described recapping and during the next 37 years, 36 publications mentioned recapping. However, since Oddie et al (2018) 40 new publications mention

recapping, of which 32 cite Oddie study (data source google scholar). Furthermore, since 2018 recapping is been introduced into mite resistant selection programs.

3. L108ff: It should be added for each parameter/ trait the number of colonies/ different locations which are used in the data set. It is too difficult to extract this from suppl. mat.

We have now added the number of single colonies and colony averages in the methods, and in the Figure legend we now mention the number of data points and number of countries they were obtained from. Any more data makes to legend excessively long and illegible. The location, author and number of colonies used for every data point is also now in the supplemental tables.

4. L131: Reproductive cycles between 2 and 3? There are other data from Fries (1996).

Indeed, Fries & Rosenkranz (1996) estimated the number of reproductive cycles as between 1.5 and 2 but assumed no mite death/loss. A later study Martin & Kemp (1997) and found between 2 to 3 reproductive cycles and this range now appears to be the expected range within the literature e.g. Rosenkranz et al (2010) and Traynor et al., 2020 and hence the one we used. However, as we state the relationship was independent of the actual number of reproductive cycles used.

5. L136 (and L260ff): "increase in the number of infertile mites as they exhaust their limited supply of eggs". Here only two papers from 1984 + 1986 are cited. To my knowledge there exist no study confirming that the lack of oocytes in the Varroa female is a reason for infertility.

The reviewer is correct so we have removed the misleading words 'which was found (Fig 2e)' and added in two new references (Alberti and Hanel, 1986; Donze et al., 1996) indicating that the number of spermatozoa is also limited, the section now reads:

The model also predicts that the increased removal of mite-infested brood will cause an increase in the number of infertile mites as they exhaust their limited supply of eggs and spermatozoa (4 refs).

L260 We have now expanded this section to make it more clear what is and is not known and now reads:

Mite infertility has long been established in resistant colonies, but whether it is caused by extrinsic or intrinsic factors is unknown (Häußermann et al., 2019). A simple explanation is that the increased brood removal leads to more mites depleting running their finite supply of 18–30 eggs (Mikityuk, 1979; Akimov and Yastrebtsov, 1984; Alberti and Hanel, 1986; Ruijter, 1987) and limited supply of spermatozoa (Alberti and Hanel, 1986; Donze et al., 1996). Infertile mites had fewer spermatozoa (Harris & Harbro, 1999), and the

number of laid eggs steadily declined in any mites performing more than two reproductive cycles (Ruijter, 1987).

6. L148: Why is DWV removed from the equation?

We have now made it clear why DWV was not included in the model as it had a set effect on the mortality of pupae. Considering we needed to increase the mortality of pupae to simulate removal we also needed to remove any effect of DWV. This is now explained on L166.

7. L158: At least in temperate regions the critical infestation rates are quite lower. Are these statements supported by studies?

This finding is irrespective of region since the mite reproduction values, number of eggs laid, development times ect are all similar in all regions. Furthermore, it is almost impossible to compare critical infestation levels in untreated vs resistant bees in temperate populations since treatment is widespread otherwise, they would die, hence the lack of data. There is some data from temperate UK resistant colonies and is not significantly different from the tropical data, so this is now mentioned in the main text L192-195

8. L177ff: These statements are not supported by literature!

We have deleted these statements and now discuss in more detail the evidence behind long-term changes, where the data exists, and as requested earlier broaden the scope to introduce some of the other ideas suggested.

9. L197: This generalized statement is only supported by one study on a small population in Southern France.

We have now provided additional support from two other resistant populations, in the UK and one in the S. Africa.

10. L200ff: This statement must be explained and justified through the results.

We have explained and justify both at the start of the discussion as suggested but also in more detail later in the re-written discussion following the advice of all 3 reviewers.

11. L 267-276: These important statements are only justified by two papers from 2002 (one dealing with AFB)

We have now obtained additional published material to support some of these statements, but also following the advice of the reviewer expanded this section to include other possibilities to explain the current data. (See L 297-310)

12. L278ff: This statement is not supported by the data or the cited literature!—

Ultimately, this study presents a potential framework to explain the common traits shared by resistant colonies using all the published data we could find. In the revision we have expanded the scope of the discussion and identify gaps and testable hypothesis where more data is needed. Only time and more targeted experiments will tell if the proposed framework remains robust or not.

13. Some important papers/aspects are missing in the discussion, for instance:

All the suggested papers below are now cited and discussed in the rewritten and expanded discussion.

Eynard et al., (2020) assuming that VSH and SMR are not linked!

The high variability in measurements for both VSH and SMR traits means that they are unlikely to correlate unless very, very large data sets are collected, and this can be determined using power analysis since Fig. 1 provides some idea of trait variation.

Variability is discussed at L221-233

Schöning et al. (2012): Removal of brood/ mites elicited by DWV symptoms.

We have included this ref in the amended mite/pupa detection section (L251-265), along with the findings from the excellent new paper Mondet et al., 2021.

Page et al. (2016, and others): "altruistic suicide" in *A. cerana* as an important trait of the original host.

This paper and other as cited in a mostly new discussion on *A. cerana* including the idea of 'altruistic suicide'. Entombing and social apoptosis (L312-330)

Journal Name: Proceedings of the Royal Society B

Journal Code: RSPB

Print ISSN: 0962-8452

Online ISSN: 1471-2954

Journal Admin Email: proceedingsb@royalsociety.org

MS Reference Number: RSPB-2020-2795

Article Status: REJECTED

MS Dryad ID: RSPB-2020-2795

MS Title: Parallel Evolution of Varroa Resistance in honey bees; a common mechanism across continents

Appendix B

Referee: 4

Comments to the Author(s).

The authors present their results of a metanalysis of 58 papers on the topic of honey bee resistance to the ectoparasitic mite *Varroa*. This approach provide data across continent all over the world and all the spectrum of beekeeping conditions including climate and environment. It turn out that all the resistant populations have associated the following 3 traits: recapping, brood removal and reduced mite reproduction. It has implication in the understanding of honey bee resistance to their parasites to eventually reduced treatments or even keep bees without the need of any treatments while maintaining performances.

This is a timely topic and would be a nice contribution to the fields. The paper is well written in a synthetic straight to the point way what in turn ease the read flow.

I have however several concerns

General comments

First of all I have concerns about the structure of the manuscript.

I was expecting a research article. As said, it is well written, in a nice English, in a clear logic. While reading I quickly realized I was more and more confused and I had more and more difficulties to understand. At the end, I have the feeling maybe it would be better to make a review? Or in a format where you can do in the same time Results in Discussion? As it stand, to my opinion it is not a structure of a Research Paper. I know there is no clear definition about a definition if a literature metanalysis better fit as Research article or Review. But nothing stand in the way of doing a metanalysis in a Review. Here you quote more than 160 papers...

We agree that the methodology used does not fit the description of a meta-analysis as pointed out by the editor and so have opted to no longer use the term meta-analysis. The majority of the text from the results has been removed to the discussion in order to fit the traditional article format.

Also you results part is partially already a discussion. It is not only about presenting the results of the framework you build but also put in perspective with the literature. SO sure you research is using papers to extract information out of it, so when you comment your results, in a way you talk about the different paper you used and you quote them. But in the results part you don't quote at all the 58 paper you used for the frame work, but 49 references from literature. It sounds a lot to me for a results section. I would expect in a Research paper you present only of the results per se, and then go to the discussion and put in perspective and quote a lot of refs there.

Agreed the results was originally more in the format of a results and discussion section as previous reviews liked this approach but addressing a large number of comments from one reviewer substantially change the manuscript. Following the advice of yourself and the editor we have altered it so that the results and discussion are now more clearly separated.

References are still quoted within the description of the framework in the results this was to explain the justification for how we connected the various traits

In addition to me, your testable hypothesis is not clear. Actually, you looked resistant population in literaute and try to found out common resistant trait. However, the way you present your aim

sounds that resistance is assumed to be fixed and established and that out of the literature it will demonstrate that the evolution fixed in population resistance traits. At the end, I see an aim but not a hypothesis, so that also there is a discrepancy with the title of the paper.

Agreed – we had aimed to compare the expression of various resistance traits between susceptible colonies and resistant colonies (those populations that had survived a long time without treatment). Using the outcome of these comparisons we aimed to create a hypothetical framework to explain them, a framework which could be tested in future studies. – this has been altered so that this is explained at the end of the introduction

We don't view resistance as a fixed trait but the product of a combination of traits, due to the known complexity of hygienic behaviour and learning to detect a new pest, but we suggest that parallel evolution may explain the repurposing of pre-existing traits to control *Varroa* irrespective of location, especially since all studies concern a single species (*A. mellifera*) against the same pest (*Varroa*).

As I mentioned earlier this is hot and timely topic. A lot was published lately. I have the feeling that many references are missing in your reference list. Starting with nice reviews that has been recently published: Le Conte, Buchler et al in *Insects*, Mondet et al., Traynor et al, Guichard et al. Actually the last one is a review that study in detail the resistance topic and that I would have expect to be mentioned at least in the discussion. Indeed there is plenty to discuss. Just one example: surviving colony or population does not necessary imply that the colony is resistant or tolerant to *Varroa*. It could also be due to environmental reasons. When you move a "resistant" colony in a new place, suddenly the colony is not anymore resistant at all...

We agree, local adaptations also relate to resistance or long-term survival. Resistance in our definition is not a fixed trait and can change if a colony is moved as resistance is an adaptation to local conditions in addition to the behaviours used.

This is now explained within the introduction:

"Similar to (Buchler et al., 2010), we view resistance as the ability of a colony or population to survive long term without chemical treatment for *Varroa* within a given environment. Thus, we don't view resistance as a fixed trait but the product of adaptive traits and adaptation to the local environment."

The references have now been added where appropriate.

There are two references by Guichard et al. that are applicable to the study and thus I am not entirely clear to which you refer. However, both discuss the difficulties in selecting for resistance using traits which is an important talking point and has been added to the discussion.

"Ultimately, variability severely affects selection programmes [reviewed in (131)], whereas, in natural selection based experiments such as bond experiments (15), black box experiments (13, 132) assumptions on the importance of traits are not made."

[Guichard, M., Dietemann, V., Neuditschko, M. et al. Advances and perspectives in selecting resistance traits against the parasitic mite *Varroa destructor* in honey bees. *Genet Sel Evol* 52, 71 \(2020\). <https://doi.org/10.1186/s12711-020-00591-1>](https://doi.org/10.1186/s12711-020-00591-1)

Additionally, you maybe might develop more the methods. It is a bit short and confusing. You might want make a figure or table to make it easier for the reader. For example you start with "Using all the current available data"...But at this point I have no idea about which data you are talking about. Then since you take paper all over the world, this indeed mean Different type of bees, Different Varroa Haplotypes, Different populations, different protocols to measure the traits (without talking about accuracy and reliability of the measurements), so in one word differents conditions. In addition, what about environmental effects? This make impossible to compare the data. That says I know you are not comparing. More putting together, aggregating data. Therefore, I would expect you validate the way you worked or explain why you believe it worked to put all of this together and might be analysed and interpreted...

We have expanded the methods to give a clearer explanation as to how we searched for and collected the data.

In the supplementary data we have added a section to explain the incidence of different varroa haplotypes and why they are unlikely to affect the results.

We have also added an explanation for each trait on how we controlled the protocols and sample sizes used.

We have also added an explanation to the introduction and discussion for our use of different populations and sub-species of bees and why the data can be combined.

In the intro

"However, three studies (27, 33, 34) using the same methods found two traits (increased recapping and mite infertility) in Varroa resistant populations in South Africa, Brazil, France, UK, Norway and Sweden, countries with different environmental conditions (tropical to sub-artic). This indicates that Varroa resistance has arisen in multiple locations, irrespective of honey bee variety or environment"

In the discussion

"Although, many local sub-species exist, *A. mellifera* remains a single species and environmental conditions within the colony i.e. those that Varroa are subject, remain remarkably constant irrespective of location, which has aided its semi-domestication and global distribution."

Finally, in your discussion, about the topic of at least 7 countries and Hygienic behavior it is not very clear to me. Which seven countries are you talking? Could you clearly mention the refs?

This sentence has since been removed from the discussion however to clarify the 7 countries referred to countries from which we had extracted the data as well as published data. So Brazil, Mexico, The Netherlands, The UK, South Africa, Germany, Sweden, Norway etc.

I do not see how you results can lead to demonstrate what you say in the discussions....

There is 30 years of research on Hygienic behavior and so far no clear link direct and indirect with Varroa resistance in the field. There is no selection program using Hygienic behavior that succeeded to improve resistance. And there is no clear link on Hygienic Behavior and survival of hives (rather negative). There is no clear correlation between Hygienic behavior and Varroa resistance in the field. (In Africa, Kenya for example, or a recent review of Spivak and Danka, in *Apidologie*, 2021, or Leclercq et al., 2017 and 2018 or Guichard et al., 2020). Alternatively, maybe I did not understand what you

meant with Hygienic Behavior. So in this case please add a clear definition. In all resistant population, High Hygienic Behavior has not been systematically demonstrated.

This is not surprising since almost all the above study measured hygienic behaviour using the freeze brood or pin kill methods. This determines the ability of bees to detect dead pupa, not infested pupa (although this was often wrongly assumed) since the vast majority of Varroa infested cells are alive, thus allowing the mites to reproduce, hence they selected for the wrong trait. This idea is further supported by Mondet et al (2021) who indicated Varroa resistant bees that detected and uncapped infested cells were unique in their ability to detect the difference between based on several ketones and acetates, as we mention in the ms. Freeze killed and pin killed brood methods are not correlated with the removal of varroa infested brood (leclercq et al., 2017). Thus, the failure to establish mite resistant lines based on these assays is understandable.

The widespread use of freeze kill brood has led to the misconception that this represents 'hygienic behaviour'. However, the removal of dead brood is only one aspect of this general trait against a whole suite of pest and pathogens of honeybee brood. Each cue (chemical/vibrational) to initiate the process of hygienic behaviour is unique, but thereafter the process is basically the same, detection, uncapping, removal/cannibalism or recapping. Therefore, as requested we have made it more clear what be consider to be hygienic behaviour.

If seen over continents, results are more about recapping results.

Of the five continents habited by *Apis mellifera* and *Varroa* we have managed to include four (Africa, South America, North America and Europe) in the results. Note we could not compare with susceptible colonies originating from Africa and south America as none have been found so far (this is mentioned in the methods). We don't have data from Asia as it is not available.

Brood removal – Europe (Germany, UK, the Netherlands) South America (Brazil), North America (USA imported bees, Mexico) Africa (Tunisia, South Africa).

Mite infertility – Europe (Austria, Yugoslavia, Sweden, the Netherlands, Greece), North America (USA, Mexico, Costa Rica) South America (Brazil), Africa (South Africa, Ethiopia)

Recapping – Europe (France, UK, Sweden, Norway, the Netherlands), South America (Brazil), Africa (South Africa), North America (Hawaii)

Detailed comments

Maybe you put too much emphasis on DWV in the abstract and Introduction for then not really talk about it in results (a bit in Honey bee behavior, and Colony level effects; but without so much details or dedicated section) and then in Discussion (but not so detailed)...

DWV is mentioned a lot within the abstract and introduction as it is a very important part of what makes Varroa lethal to bees. We believe it is important when discussing Varroa resistance that the relationship between Varroa, DWV and bees is explained in full. Particularly, because in the absence of DWV Varroa infested colonies survive without any treatment, e.g. Fernando de Noronha.

Sometime you say framework and sometime metanalysis- Maybe I misunderstood...maybe stick with one? I would say metanalysis?

Agreed – this was confusing, as we have not followed the typical conventions of a meta-analysis it will now be referred to as a framework

I would suggest to well establish definitions. For example: Until you say in method resistance is surviving more than 5 years, I thought it meant reducing the number of parasite in the population.

Following previous studies such as buchler et al 2010 we utilised long term survival as the definition for resistance, as does COLOSS - This has been clarified in the introduction and methods.

I see that you avoid common traits name such VSH or SMR or more recently MNR. Which is actually a good idea. Because sometime people use the word of VSH for SMR or vice versa etc. And there is still not really a standard terminology on the field....So indeed it is a good idea to be more general and saying like you do brood removal or infertility, or reduced mite reproduction. I would say they is room for improvement: maybe you could try to better clarify your terminology/nomenclature and make it clear in the text all your definitions and how the reader should understand it.

We fully agree as these terms are causing much confusing. Anyway we have added detailed definitions we used for each trait within the methods and have also included simplified definitions within the introduction and the results

L39: what is “this”. What does it refer to?. Not clear. Specify. The bees naivety e.g. never exposed to Varroa means that they have not developed mechanisms to control the mite and so mite populations could grow quickly within colonies

L45: interesting and important point. I would expect more refs here.

More references added

L48: Maybe this is not that easy as it sound. Here I disagree. What about haplotype? Here in Fernando de Noronha we are talking about Japanese Haplotype. And probably this might be the only population known with Varroa Tolerance.....

The resistant-haplotype was an old argument used to initially explain the Varroa resistance of Africanised honeybees then later the Fernando de Noronha population. This argument didn't hold for Africanised bees that remain resistant with the K-haplotype and Brettell & Martin (2017) compared the mite's reproductive traits on Fernando de Noronha (J-type) with those of the K-type using the same method and found no differences. Unfortunately, we were unaware of the role of recapping at that time so it was not measured, but the aim is to do this as soon as a bee research can visit the island which is very difficult especially for research. In the abstract we have made it very clear it is the Varroa-DWV association we are concerned with and the resistant mechanism in the 3 viral free populations may be the same or different, but this is of little of no relevance to beekeepers.

L58: I do not fully get why you distinguish Europe from Uk. Geographically speaking this is the same region, and might probably all together be described as Europe. Why not say two regions (one side of the Atlantic, and the other Side with USA).

We separated the UK from Europe as it is a separate landmass and also to prevent any confusion given the current contention over the UKs relationship to Europe.

L61. Which one are you talking about? Not clear here. I don't understand what are those 7 populations....

Sorry, this was not clarified, here we are referring to the seven sub species that have featured within our data sets. This has now been clarified in the text

L.68: based on the introduction it is not clear that it arised. It does sound like a fact. But is that really sure they are truly resistant? If so, when moved in a new locations, should those populations still not be resistant? In it is not necessary the cases....

Typically, resistance does involve environmental interactions as colonies do adapt to their local environments so moving a resistant colony to a new area can cause the colony to collapse, due to a large influx of mites for a nearby feral of managed colony. We discuss this in the discussion section colony level effects, a sudden influx of mites can cause even a resistant colony to collapse as resistance traits can be detrimental if over-expressed.

Result section: Could you precise: was the 3 traits always in all the resistant populations? Or was that one of them? Which one? This is very important, because so far I do not know any "universal" traits that work in any populations.

These three traits are now presented and described in the introduction. The three traits are brood removal, recapping and mite infertility. These were chosen as they are consistently reported in long term surviving colonies and there are numerous studies that indicate this (Oddie et al., 2018, Locke et al., 2011, Locke et al., 2012, Martin et al., 2019, Hawkins & Martin, 2021)

The results have been altered, now we present our findings for the three traits (as well as their effects on populations) before putting together the framework.

L240: I am unsure about the word "demonstrate". Maybe reformulate with something like suggests. Or if you want underline then maybe "shows".

The sentence has been removed during the reformulation of the discussion – instead we discuss our proposed framework and how it centres on the idea that hygienic behaviour (mite infested brood removal) can explain the other behaviours. The hypothetical nature of this framework has been emphasised

L274: How do you know the Freeze killed is the most commonly used method for hygienic behavior? And where? In Europe, US, where else or the World?

In Leclerq et al., 2017 and le conte 2020 they mention how fkb or pkb is a routinely used test for hygienic behaviour. The sentence has been edited to include the reference and also to describe fkb as typically used test rather than the most common

L275: this is maybe a style issue, and each one his own style. But I would not expect to see "don't" in a scientific publication but rather "do not" in full word

We have amended it 'do not'

L307: the paper quoted here start to be a little bit old. Is there anything newest as well? And it deals

mainly with VSH. So do they really looked at removing and recapping? Did they see it once? Or in only one population? Or a lot? I mean maybe it is hard to generalize, extrapolate or effent say this sentence based on one paper, or?

There are few papers that have looked at the removal of drone brood and these tend to be old papers. It has been observed by multiple researchers that drone brood is not removed or recapped which is likely why it hasn't been studied in detail more recently.

L 316. Why is jacobsoni between brackets?

It is in brackets as the sentence mostly refers to Varroa jacobsoni but Varroa destructor is found on Apis cerana in China and similarly avoids worker brood.

L337: Like line 275, I would expect full wording instead of "won't"

Agreed – amended

L338: well I am not sure is it possible to say "same solution"...Or at least on my opinion your data are not sufficient to say that. You might want maybe reformulate...

Solution has been changed to "same three traits arising in populations in different continents" as we found with our data the presence of these three traits in resistant populations across different continents.

References

Look good! I did not see any problems or typos...well done

Supplementary data:

It is well prepared and presented.

Well done.

Appendix C

1) you state that you have conducted a meta-analysis. However, there are very clear methods for such an analysis and you have not followed them (see <http://www.prisma-statement.org> and <https://doi.org/10.1038/s41559-020-01295-x>). In particular, you do not report how you searched the literature (search terms, engine, date, papers collected and rejected at each stage, etc.), and your statistical methodology does not hold up from a meta-analysis perspective (you do not account for sample sizes, use Hedges' D to look at effect sizes, etc. etc.). You either need to report all of this information and do correct statistical analyses, or avoid using the term 'meta-analysis'. Either way, you still need to report how you searched for and screened the papers you base your analyses and model on - without this your study is not repeatable, and if it's not repeatable then it is not good science

We agree that the methodology used does not fit the description of a meta-analysis and so have opted to no longer use the term meta-analysis. The methods have been updated to include the search terms, engine, date, and selection criteria. We have also changed the analyses of the data by accounting for sample sizes in generating the averages.

2) your results come in two forms (i) the statistical analyses of the datasets you have gathered, and (ii) a model putting all of these together which acts as a hypothesis to explain resistance to Varroa. At the moment, the way these are presented is confusing. You put the model first as a figure (with no preceding text) and then your results section bounces back and forth between the actual results and the model that you put together from them. This is not helpful - indeed, is actually confusing - to a reader. I would suggest that this section needs complete restructuring, to present the analyses of published data first, and then the construction of the model, with explicit acknowledgement that the model is a hypothesis

We agree – the results have now been streamlined to contain only the statistical analysis and presented data. The majority of the text has been moved to the discussion. Additionally, we have moved the hypothetical framework to the end of the results to show clearly how we construct it.

3) the justification for the structure of the model is lacking. Given that this is the key outcome of your study, the justification needs to be clear and explicit. Why is this the only, or the best way, to put the data together and interpret them? You're basically asking your reader to take all of this on faith. As an example, lines 202-203 make a huge factual claim, but it is actually an interpretation of the data not a fact.

We agree our hypothesis is an interpretation of the data, this has now been explicitly mentioned within the results. We have also included several paragraphs detailing why we made the links between each trait to build the framework

4) significant sections of the Results should be in the Discussion (e.g., lines 615-630)

We agree – the results have now been streamlined to contain only the statistical analysis and presented data. The majority of the text has been moved to the discussion.

5) the Discussion starts far too strongly - the sentence in lines 1308-1308 is a much more measured statement of what you have found. Fundamentally, you have analysed data, shown differences between resistant and susceptible populations, and created a model framework from this to explain resistance that now needs to be explicitly tested. There is real value to this, but this is not how you

present your work.

The discussion was indeed too strong for a hypothetical framework, this has been addressed and the fact that this is hypothetical has been expressed more clearly.

In addition to these issues, the points made by the 4th reviewer (some of which overlap with my comments above) need addressing. I hope you find this useful as you revise your manuscript.

Appendix D

Referee: 4

Comments to the Author(s).

The authors take advantages of published data of 60 publications to better understand Varroa resistance evolution in *Apis mellifera*. They compare susceptible colonies and resistant, focusing on 3 phenotypes Brood removal, recapping and mite infertility. Based on the results they constructed a Framework to link the traits and deduce mechanisms. The knowledge pave the way to enhance selection efficiency in breeding programs towards Varroa resistant colonies.

This an interesting work, that scientifically sounds.

Thank you to the authors for their work.

General comments

This is a timely topics that indeed need to study in deep to offer solution in the field in a near future. The results here stimulate thinking, and the ideas discussed here bring fuel to the debate. I think the paper is clear, correctly build and straight forward. Sure there is convergence in many place in the world, but still on my opinion, it stay at this stage in this paper correlative results, and not prove of mechanisms. So that the concepts and idea brought here are nice, but still need to be proven in field.

Detail comments:

L14: I do see the S at the end of populations. Still this sound that many population around the globe are surviving, the majority. But still probably, at least in the managed colony with western honey bee in occidental country, the vast majority need treatments. Only the minority is surviving....

It is hard to quantify the number that are surviving as there is less data however especially concerning feral colonies which according to many uk beekeepers are returning, but this is often only speculation. You are probably correct that the number requiring treatment is greater than those surviving. We now have good data from 2 independent UK surveys that indicate around 25-27% of managed bee colonies are been untreated and this proportion is increasing but as this is not yet published we have changed it to small but increasing number of populations.

L 19: I would suggest to remove “ for the first time”. To write this does not make your MS any better.

This has been removed

L24: Not sure what is Natural Varroa Resistance here. It come then only one time in the text about Fernando de Noronha, but that’s it. Not definition. But you define in the other hand resistant population L 92. Is that the same? I believe so...Here definition is important since the reader may think that resistance bee are colonies that have load Varroa load and survive, in opposition of Tolerance with high varroa load.

Amended – our definition of natural resistance was resistance that had occurred without purposeful selection having occurred. However, we have decided to just use the term resistance for clarity so was accidentally left in from a previous version and has now been omitted.

L 55: probably I would expect to fully write the word and not “don’t”. amended

L.90: Actually why directly focuses on this 3 specific traits. Not very clear to me. I would expect that you explore all possible Varroa phenotype, only then you see the 3 best, or the one that are the most important, or in common in all populations....

The three key traits we selected are those that have a wealth of scientific backing and data. Brood removal has long been heralded as a key trait in varroa resistance as has mite non reproduction. There are several iterations of brood removal behaviour including VSH, hygienic behaviour etc however we believe it to be one key phenotype. The newer trait recapping has less history however, since the breakthrough article by oddie et al there has been increasing in evidence for it. We could have taken this approach if all Varroa phenotypes have been studied to similar levels, but there has been dozens of ideas why bees are resistance and it would have been a very lengthy ms if we had taken this approach, so focused on the three most consistently reported key traits.

L105-106: Probably fine to leave it here, but it could be also place in the results part. – we thank you for the suggestion, however, after trying it in the results it was decided that it was better placed in the methods

L135 – 136_ Pre or Per? Amended – per

Results

Throughout the results part and also later in the article, you talk about figure 1 . But then it is figure 8 in the Line 216

Amended – it is now figure 1 in L 216

Discussion:

Globally it is a nice discussion.

The beginning maybe does not fit very well with the goal of l 79-83. I was expecting something like all population have in common three traits and you will discussed that here. But I had the feeling reading the first paragraph that you would like to show how Varroa resistance arise in those population. And you emphasize the framework. Make sense to discuss about it. I might have misunderstood, but my understanding is that it is not the discussion of the results per se, but more your vision. Sure it is an attempt. But maybe I would have expect the discussion about that and speculation later in the discussion part of the manuscript.

The discussion is structured as such in order to appropriately explain how the results we obtained led us to create the hypothetical framework. The framework is our vision but also a major result of the paper and as such we were required to justify how this result came about.

After reviewing your comment and the discussion we have decided to add a sentence to highlight that we found the traits to be common amongst resistant populations.

L229 - “Here we found that the enhanced expression of these three key traits is common amongst resistant populations”